# Two Heads Are Better Than One: A Multi-Agent System Has the Potential to Improve Scientific Idea Generation

## Abstract

The rapid advancement of scientific progress requires innovative tools that can accelerate discovery. While recent AI methods, particularly large language models (LLMs), have shown promise in tasks such as hypothesis generation and experimental design, they fall short in replicating the collaborative nature of real-world scientific practices, where diverse teams of experts work together to tackle complex problems. To address the limitation, we propose an LLM-based multi-agent system, i.e., Virtual Scientists (VIRSCI), designed to mimic the teamwork inherent in scientific research. VIRSCI organizes a team of agents to collaboratively generate, evaluate, and refine research ideas. Through comprehensive experiments, we demonstrate that this multi-agent approach outperforms the state-of-the-art method in producing novel and impactful scientific ideas, showing potential in aligning with key insights in the Science of Science field. Our findings suggest that integrating collaborative agents can lead to more innovative scientific outputs, offering a robust system for autonomous scientific discovery.

## 1 Introduction

The rapid acceleration of scientific progress necessitates more efficient and innovative tools for exploring new concepts and tackling complex challenges (Park et al., 2023b). The concept of automatic scientific discovery has emerged as a promising solution to expedite innovation, representing a long-standing ultimate goal within the scientific community (Langley, 1987). With the development of artificial intelligence (AI), automatic scientific discovery has witnessed the potential to revolutionize how research is conducted by automating key steps in the scientific process, ranging from hypothesis generation to experimental design (Raghu & Schmidt, 2020; Spangler et al., 2014).

More recently, foundational models, especially the large language models (LLMs) (OpenAI, 2023; Dubey et al., 2024), have shown significant progress in general capabilities, facilitating their applications in various stages of scientific discovery (Abramson et al., 2024; Chang & Ye, 2024), including literature reviews (Hsu et al., 2024), experimental designs (Huang et al., 2024), *etc*. A notable development is the AI Scientist (Lu et al., 2024), which introduces a scalable system for end-to-end scientific paper generation, highlighting the potential of LLMs to drive autonomous scientific discovery. Despite its capabilities, AI Scientist operates with ONE agent, which falls short of replicating real-world scientific practices, where research is often conducted through collaborative efforts involving diverse teams of experts (Kayacik et al., 2019).

To address the limitations of a single executive system, such as the AI Scientist, and to better replicate the collaborative nature of real-world scientific discovery (Gauch, 2003), we focus on the idea generation phase in the research process, which demonstrates more collaborative aspects (Linsey et al., 2005). From this end, we propose an LLM-based multi-agent system, Virtual Scientists (VIRSCI), designed to harness the potential of LLM agents in assisting autonomous scientific idea generation. Leveraging the inherent human-like reasoning capabilities of LLMs (Xie et al., 2024), VIRSCI simulates the collaborative process of research idea generation, which is divided into five steps (see Fig. 1): (1) *Collaborator Selection*, (2) *Topic Selection*, (3) *Idea Generation*, (4) *Idea Novelty Assessment*, and (5) *Abstract Generation*. To be more specific, we construct a knowledge bank of the background of scientists of interest and develop digital twin (Tao et al., 2018) agents

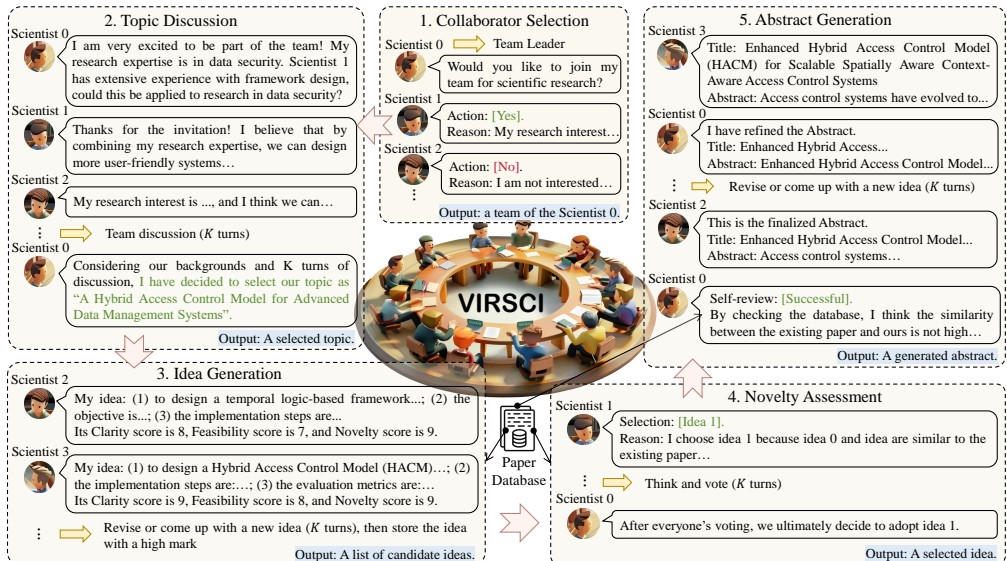

Figure 1: The proposed LLM-based multi-agent system, VIRSCI, consists of five key steps: **Collaborator Selection**, where a research team is assembled; **Topic Discussion**, where the research topic is determined; **Idea Generation**, where team members propose and refine ideas; **Novelty Assessment**, where ideas are evaluated and voted on to select the best one; and **Abstract Generation**, where the selected idea is developed into a complete abstract.

of real scientists using a retrieval-augmented generation (RAG) framework (Gao et al., 2023). The lead agent, or team leader, identifies appropriate collaborators based on the scientist collaboration network, aligning their expertise and research interests to reflect real-world cooperation patterns. Using a past paper database, the team retrieves reference works to guide the formation of novel ideas. Moreover, we implement a "**team discussion**" mechanism in subsequent stages, where collaborators engage in iterative inter- and intra-refinement dialogues, to enhance the quality of each output. Finally, the team generates a comprehensive abstract representing the proposed ideas. To evaluate the novelty of the ideas, we introduce a benchmark for the measurement from three perspectives: dissimilarity to past papers, alignment with contemporary research trends, and the potential influence in contemporary research (Shao et al., 2020; Yang et al., 2022). By comparing the abstract against both a past and a contemporary paper database, we ensure the generated ideas are innovative while aligning with emerging scientific directions, thus validating the effectiveness of our approach.

We conduct comprehensive experiments to verify the effectiveness of VIRSCI in producing novel scientific ideas. The findings demonstrate that the multi-agent system outperforms the single-agent executive with an average gain of 13.8% and 44.1% in alignment level and potential impact on contemporary research, respectively. Additionally, our experiments reveal emergent social behaviors among agents, aligning with prior studies in Science of Science (Fortunato et al., 2018; Wu et al., 2019; Zeng et al., 2021; Shi & Evans, 2023), suggesting the potential for further exploration of the mechanisms in research collaboration using multi-agent simulations. To sum up, our core contributions are summarized as follows:

- To the best of our knowledge, we propose the first multi-agent system for conducting scientific collaborations in an end-to-end pipeline from team organization to novel scientific idea generation. Furthermore, the real data is utilized for role-play and the objective evaluation of final outputs.

- We conduct extensive evaluations to investigate VIRSCI in terms of the team settings and the novelty of generated scientific ideas. The results demonstrate that multi-agent collaboration can improve the quality of the outcomes, surpassing the SOTA single-agent method.

- The simulation results align with the important findings in Science of Science, such as fresh teams tend to create more innovative research, showcasing the potential of VIRSCI as a powerful tool for future research in this field.

## 2    RELATED WORK

### 2.1    AI FOR SCIENTIFIC DISCOVERY

In recent years, AI has fundamentally reshaped the landscape of scientific discovery by providing powerful tools that enhance various research processes (Xu et al., 2021). AI techniques, especially generative AI, can facilitate basic scientific discoveries, such as identifying complex molecular (Vignac et al., 2022) and protein structures (Abramson et al., 2024), drastically reducing the time required for experimental iterations. These advancements have found wide application across diverse fields such as chemistry (Liu et al., 2023a), meteorology (Bi et al., 2023), and medicine (Rajpurkar et al., 2022), *etc*. Besides, with the advent of LLMs, AI methodologies can step further and collaborate in streamlining critical stages of the scientific pipeline, including hypothesis generation, experimental design, data acquisition, and analysis(Zheng et al., 2023; Wang et al., 2023; Miret & Krishnan, 2024; Wysocki et al., 2024; Lu et al., 2024). Nevertheless, these approaches lack the collaborative nature of the scientists intrinsic to real-world research. VIRSCI is the first to harness the power of an LLM-based multi-agent system to facilitate the generation of research ideas in autonomous scientific discovery.

### 2.2    MULTI-AGENT SYSTEMS IN TEAM COLLABORATION

A multi-agent system for team collaboration leverages autonomous agents to coordinate, communicate, and solve tasks within a shared environment, mimicking human teamwork dynamics (Dorri et al., 2018). Traditional multi-agent systems typically involve semi-autonomous agents coordinating through explicit protocols and structured messages to achieve common goals (Dunin-Keplicz & Verbrugge, 2011; Bakliwal et al., 2018). The advent of LLMs has revolutionized this landscape by enabling agents to utilize natural language for communication and collaboration in a believable proxy of human behavior (Park et al., 2023a), thereby fostering a more intuitive and flexible interaction model. Recent studies have further verified the superior performance of LLM-based multi-agent systems in various domains, such as programming, game playing, and complex reasoning tasks when compared to single-agent execution (Liu et al., 2023b; Wang et al., 2024; Light et al., 2023; Du et al., 2024). In this work, we strategically implement the power of LLM-based multi-agent systems to function as collaborative scientists, promoting *de novo* scientific ideas.

## 3    THE VIRTUAL SCIENTISTS

In this paper, we aim to build a multi-agent system using a real-world academic dataset to simulate how a scientist assembles a research team and collaboratively generates an abstract that details a novel scientific idea. Our VIRSCI system consists of two components: a scientific research ecosystem and a multi-agent system for scientific idea generation.

### 3.1    THE SCIENTIFIC RESEARCH ECOSYSTEM

The scientific research ecosystem comprises two main components: paper information and corresponding author information ranging from year $y_{start}$ to $y_{end}$. First, we select a year $y_{bound}$ as a time point and split the papers into two subsets: past papers $B_{past}$ and contemporary papers $B_{con}$. We further extract authors from $B_{past}$ to form the complete set of scientists $S$, with each scientist's background information stored in the author knowledge bank, and the adjacency matrix $A$, which represents the collaboration counts between scientists.

**Past Paper Database** To construct the past paper database $B_{past}$ using the Faiss, we selected papers published before the $y_{bound}$. Each paper includes essential information such as its title, citation count, and abstract.

**Contemporary Paper Database** The contemporary paper database $B_{con}$, also constructed with Faiss, consists of papers published after $y_{bound}$. Similarly, each paper's basic information is structured in the same way as the past papers. Although using papers from this time range may raise concerns about data leakage, given that LLMs are trained on data within this period, we will explain in detail why this does not pose a threat to the overall validity of our experiments and conclusions in the Appx. C.

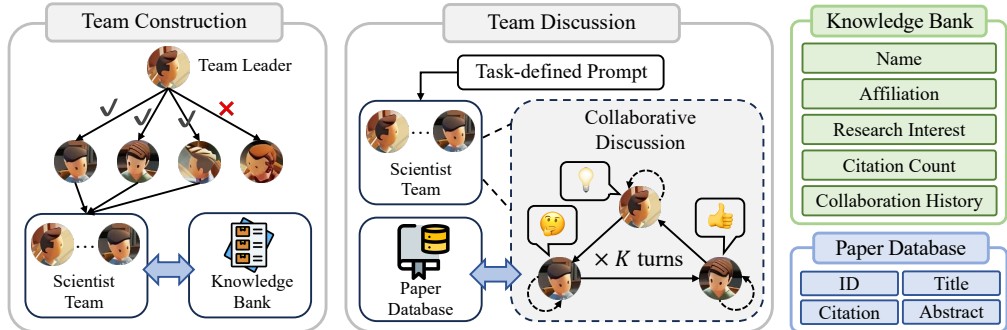

Figure 2: Key components of the proposed system. The left section illustrates the collaborator selection process, where the team leader forms a research team. The middle section highlights the discussion routine, a fundamental part of every step in the system, where the team engages in collaborative dialogue to progress through tasks. The right section depicts the architecture of the author knowledge bank and paper database, which provide critical information used throughout the collaboration process.

**Author Knowledge Bank** For each scientist in $S$, we extract their basic profile from the computer science dataset, which includes their name, affiliations, citation count, research interests, and collaboration history. Using the KnowledgeBank module from AgentScope(Gao et al., 2024), we embed these scientist profiles into the author knowledge bank. This allows agents to quickly access and familiarize themselves with other initialized agents' information. Notably, real author names are masked to prevent data leakage and privacy problems during agent initialization (See Appx. B).

**Adjacency Matrix** Given the scientist set $S$, let $A$ represent the adjacency matrix, where $A_{i,j}$ denotes the number of times scientist $i$ has collaborated with scientist $j$. To prevent agents from always choosing previously collaborated scientists, overlooking the benefit of fresh collaborations that often lead to more original and impactful research (Zeng et al., 2021), we increment all values in $A$ by 1. This adjustment ensures that scientists with no prior collaborations still have a chance of being selected, encouraging agents to explore new partnerships.

## 3.2 THE MULTI-AGENT SYSTEM FOR SCIENTIFIC COLLABORATION

We first randomly sample a scientist $s_0$ from $S$ as the team leader. The team leader then follows these steps to produce an abstract: (1) Collaborator Selection, (2) Topic Discussion, (3) Idea Generation, (4) Idea Novelty Assessment, and (5) Abstract Generation. To help each agent become familiar with the backgrounds of other team members without overloading the initialization prompt, we employ retrieval-augmented generation (RAG) (Lewis et al., 2020), used throughout all five steps. All necessary prompts and example scenarios are shown in Appx. F and G.

**Collaborator Selection** The first step in our system is collaborator selection, aimed at forming a team of scientists, $T = \{s_0, \ldots, s_i, \ldots, s_n\}$, where $n$ denotes the team size. When $s_0$ is selecting collaborators, we convert the adjacency matrix, $A$, into a probability distribution using the following equation: $P_{i,j} = \frac{A_{i,j}}{\sum_{j=1}^{N} A_{i,j}}$, where $N$ denotes the size of $S$. This allows the team leader to iteratively send invitations to preferred collaborators. Upon receiving an invitation, the invited scientist evaluates whether to join the team using the chain-of-thought process (Wei et al., 2022), considering the profiles of $s_0$ and the current team members. If accepted, the scientist is added to the team $T$. This process continues until the pre-defined team size $n$ is reached.

**Topic Discussion** The next step is to propose a research topic, which will guide the research direction. Inspired by multi-round collaboration (Mezirow, 2003; Sunstein, 2005; Amgoud & Prade, 2009) and multi-agent collaboration strategies (Xu et al., 2023; Zhang et al., 2023; Shinn et al., 2024), we design a general team discussion mechanism. In this mechanism, team members engage in discussions based on a specific task description prompt. This process is applied not only to topic discussion but also to subsequent collaboration steps. While allowing agents to decide when to stop the discussion would better reflect real-world scenarios, fixing the number of turns ensures

consistent inference costs across different team settings in our experiments. Therefore, we leave the discussion of adaptive turn numbers to the ablation study (See Sec. 4.4). Given the team $T$, the prompt during the topic discussion is:

$$Q_{k,i} = \langle Q_{team}, Q_{topic}, \bigcup_{t=1}^{k-1}(\overline{D_t}), \bigcup_{j=0}^{i-1}(R_{k,j})\rangle, \quad (1)$$

where $Q_{team}$ denotes the description of the current team members, $Q_{topic}$ represents the task description for the topic discussion, $R_{k,j}$ is the response of agent $j$ at turn $k$, and $\overline{(D_t)}$ is the team leader's summary of dialogues from turn $t$, where $D_t = \{R_{t,0}, R_{t,1}, \ldots, R_{t,n}\}$. Given the prompt $Q_{k,i}$, each scientist agent generates a response $R_{k,i}$, sampled from a probability distribution $R_{k,i} \sim P_{s_i}(\cdot|Q_{k,i})$. Since agents can use RAG to access the author knowledge bank during discussions, they may seek advice from scientists who are relevant to the topic but not part of the team. In such cases, we initialize a new agent with the mentioned scientist's profile and include their responses in the discussion. However, to maintain the fixed team size, this agent is not added to the team. This process is termed the "Invitation Mechanism" and is also applied in subsequent steps, with its effectiveness demonstrated in the ablation study. An example scenario is shown in Appx. G.2.2. After $K$ turns of discussion, the team leader generates the final research topic $R_{topic}$ based on the content: $\langle Q_{topic}, \bigcup_{t=1}^{K-1}(\overline{D_t}), \bigcup_{j=0}^{n}(R_{K,j})\rangle$.

**Idea Generation** Third, the team is tasked with proposing several potential ideas. To align with genuine research workflows and mitigate LLM illusions (Huang et al., 2023), each agent is required to generate a comprehensive response that includes three key components: (1) a description of the idea, (2) a specific experimental plan, and (3) a self-assessment covering metrics such as novelty, feasibility, and clarity, representing the agent's confidence (See Appnx. 12).

At the start of the idea generation process, when no ideas have yet been proposed, the agent is provided with references by searching $B_{past}$ using the topic $R_{topic}$, denoted as $B_{past}(R_{topic})$. The first idea-generation prompt is defined as:

$$Q_{1,0} = \langle Q_{idea}, R_{topic}, B_{past}(R_{topic})\rangle, \quad (2)$$

where $Q_{idea}$ represents the task description. Inspired by the concept of gradually expanding an archive of ideas (Zhang et al., 2023; Lu et al., 2024), when a scientist $s_i$ at turn $k$ receives an existing idea from the previous response $R_{k,i-1}$, we retain the previously generated ideas along with their corresponding references from $B_{past}$. These are passed to the next agent, who can either refine the existing idea or propose a new one, depending on its choice. The prompt is represented as:

$$Q_{k,i} = \langle Q_{idea}, R_{topic}, B_{past}(R_{k,i-1}), \bigcup_{t=1}^{k-1}(\overline{D_t}), \bigcup_{j=0}^{i-1}(R_{k,j})\rangle. \quad (3)$$

Afterwards, the response of $S_i$ at turn $k$ can be represented as $R_{k,i} \sim P_{s_i}(\cdot|Q_{k,i})$. After $K$ turns of discussion, we retain the three ideas with the highest confidence and store them in the idea list $I$.

**Novelty Assessment** To enhance the quality of ideas and mitigate agent overconfidence, we introduce an idea novelty assessment, enabling agents to compare each idea with related papers from $B_{past}$ and vote for the idea they consider most novel. Given the idea list $I$, agents search for related papers using each idea's description to determine whether it significantly overlaps with existing works. To simulate a blind review process, no dialogue memory is included in the prompt. The prompt for $s_i$ at turn $k$ is defined as:

$$Q_{k,i} = \langle Q_{check}, \bigcup_{j=1}^{3}(I_j, B_{past}(I_j))\rangle, \quad (4)$$

where $I_j$ is the $j$-th idea in $I$. Following the chain-of-thought process, the response $R_{k,i} \sim P_{s_i}(\cdot|Q_{k,i})$ includes the scientist's preferred idea and the reasoning behind their choice. The idea receiving the highest number of votes is then selected as the final idea, $R_{idea}$, for abstract generation.

**Abstract Generation** Lastly, the team is required to produce a comprehensive abstract that includes the following sections: (1) Introduction, (2) Objective, (3) Methods, (4) Expected Results, and (5)

Conclusion (Alexandrov & Hennerici, 2007). At the start of abstract generation, the team leader provides an initial draft based on $R_{idea}$. The first abstract-generation prompt is:

$$Q_{1,0} = \langle Q_{abstract}, R_{idea} \rangle, \tag{5}$$

where $Q_{abstract}$ represents the task description and format requirements.

When an abstract is provided by the previous response $R_{k,i-1}$, the next scientist's response should include: (1) an evaluation of the prior abstract (evaluation metrics are detailed in Appx. 15), (2) proposed modifications, and (3) the revised abstract to enable continuous refinement. The corresponding prompt is:

$$Q_{k,i} = \langle Q_{abstract}, Q_{judgement}, R_{k,i-1} \rangle, \tag{6}$$

where $Q_{judgement}$ is the prompt that asks agents to evaluate the previous abstract. Dialogue history is not included in this prompt since the process is iterative and focuses on refining a single abstract. Including previous versions would make the prompt redundant. After $K$ turns of revision, the final abstract is denoted as $R_{abstract}$.

A self-review mechanism is also considered after $R_{abstract}$ is finalized to pre-check its novelty. The optimized abstract $R_{abstract}$ is provided to the team leader to assess novelty by comparing it to similar papers in $B_{past}$, where more details are shown in Appx. D.1. Because it introduces uncertainty in total inference cost, making it difficult to ensure fair experimental comparisons, we only discuss the effectiveness of this module in the ablation study (see Sec. 4.4).

## 4 EMPIRICAL STUDY

### 4.1 EXPERIMENTAL SETTINGS

**Dataset** We build our scientific research ecosystem using real scientists' information from the AMiner Computer Science Dataset [1], which was constructed by extracting scientists' profiles from online web databases (Tang et al., 2008). This dataset includes 1,712,433 authors and 2,092,356 papers, covering the period from 1948 to 2014, with disambiguated author names. To manage the large volume of data, we set $y_{start}$, $y_{bound}$, and $y_{end}$ to 2000, 2010, and 2014, respectively. For quality assurance, we filtered out past papers lacking abstracts or with fewer than 10 citations, contemporary papers with fewer than 5 citations or missing abstracts, and authors with fewer than 50 papers or 50 co-authors. As a result, we extracted detailed information from 156 authors and 85,217 papers to construct the ecosystem and initialize the corresponding agents for the simulation. All paper and author data are embedded using the "mxbai-embed-large" model (Lee et al., 2024).

**Implementation** We implement our system on top of the Agentscope framework (Gao et al., 2024), which serves for LLM-empowered multi-agent applications. We evaluate our system using different publicly available LLMs: GPT-4o (OpenAI, 2023) and Llama-3.1 (8b and 70b) (Dubey et al., 2024). GPT-4o is accessible exclusively via a public API, while the Llama-3.1 models are open-weight and invoked using the Ollama (Ollama, 2024) in our experiments. Each experimental run on Llama-3.1 (8b) takes approximately 10 minutes on 1 NVIDIA A100 40G GPU within a team discussion setting of 4 members and 5 turns ($K = 5$). All experimental results are averaged on 20 runs.

**Evaluation Metrics** Since no single evaluation metric perfectly captures the novelty of scientific outputs, we employ three common metrics that align with our intuition: (1) **Historical Dissimilarity (HD)**: The average Euclidean distance between the generated abstract embedding and embeddings of the 5 most similar abstracts in $B_{past}$ (Shao et al., 2020; Zhou et al., 2024). A larger distance indicates that the generated abstract is more dissimilar from existing papers, suggesting a higher likelihood of novelty. (2) **Contemporary Dissimilarity (CD)**: The average Euclidean distance between the generated abstract embedding and embeddings of the top 5 most similar abstracts in $B_{con}$. A smaller distance indicates greater similarity to newer papers, also suggesting a higher likelihood of novelty. (3) **Contemporary Impact (CI)**: The average citation count of the top 5 most similar abstracts in $B_{con}$ (Yang et al., 2022). A higher citation count suggests that the generated abstract is more likely to have a higher impact. To ensure comparability, we normalize each calculated metric using the mean value derived from the entire corresponding database, with normalization defined as the metric divided by the mean value. Since novelty is difficult to measure directly, we introduce a

---

[1] https://www.aminer.cn/aminernetwork

proxy metric to comprehensively account for the three indicators: (4) **Overall Novelty (ON)**. We assume that ON is positively related to both HD and CI and negatively related to CD, calculated as $ON = (HD \times CI)/CD$. Mathematically, the expected value of ON is proportional to the true novelty. Further discussion of this metric is provided in the Appx. E.

**Experimental Design** We aim to explore several key aspects of VIRSCI's performance. (1) How do the evaluation metrics of abstracts from VIRSCI outperform those generated by the SOTA single-agent method AI Scientist under similar conditions (Sec. 4.2)? (2) How do the novelty scores of abstracts improve in relation to different team settings, including team size and discussion turns (Sec. 4.3.1), team freshness (Sec. 4.3.2), and team research diversity (Sec. 4.3.3)? (3) How do different components in our system affect the novelty of the generated abstracts (Sec. 4.4)? (4) How do adaptive turn numbers affect the novelty of the generated abstracts (Sec.4.4)?

## 4.2 COMPARISONS WITH AI SCIENTIST

Table 1: Comparisons with AI Scientist. Results show that our multi-agent system outperforms the AI Scientist across all metrics, with GPT-4o achieving the highest performance.

| Agent Model | Method | LLM Review Score ↑ | CD ↓ | CI ↑ |
|---|---|---|---|---|
| GPT-4o[1] | AI Scientist | 3.10 | 0.38 | 3.22 |
| | Ours | 3.34 (+0.24) | 0.34 (-0.04) | 3.78 (+0.58) |
| LLaMA3.1-8b | AI Scientist | 2.09 | 0.49 | 2.12 |
| | Ours | 2.31 (+0.22) | 0.42 (-0.07) | 3.29 (+1.17) |
| LLaMA3.1-70b | AI Scientist | 2.24 | 0.48 | 2.11 |
| | Ours | 2.53 (+0.29) | 0.40 (-0.08) | 3.36 (+1.25) |

[1] GPT-4o API is "gpt-4o-2024-08-06".

**Adjusting Settings for Fair Comparison with AI Scientist** Given that the problem formulation of the AI Scientist differs from ours, we must make several justifications to ensure relative fairness in our comparisons: (1) Since the AI Scientist is limited to generating ideas from its pre-defined topics (2D Diffusion, NanoGPT, and Grokking), we include NanoGPT in the topic selection prompt for VIRSCI as the initial discussion topic, ensuring that the final abstracts align with the same research direction. (2) Since our teamwork approach differs from the AI Scientist's solitary investigation, we need to ensure that the comparisons are made under the same inference cost. The AI Scientist performs 50 turns of self-reflection during its idea generation, which does not apply to its paper generation. To align the inference costs, we set the number of team members to 4 and the number of discussion turns to 5, ensuring the experiments are conducted under approximately the same computational cost. (3) Since the AI Scientist lacks a scientific research ecosystem, it retrieves papers across all time ranges through the Semantic Scholar API (Fricke, 2018). To maintain consistency with the AI Scientist, we replace our databases with the Semantic Scholar API for both paper retrieval and metric calculation. Specifically, after generating ideas and corresponding abstracts from both the AI Scientist and our system, we use the generated ideas as queries to retrieve related papers, extracting corresponding abstracts and citation counts for evaluation. (4) We evaluate the generated abstracts using both our metrics (CD and CI) and the AI Scientist's metric (LLM review score) (Lu et al., 2024). The LLM review score is calculated by GPT-4o which conducts abstract reviews based on a truncated version of the Neural Information Processing Systems (NeurIPS) conference review guidelines[2], shown in Fig. 17.

**Experimental Results and Analysis** As shown in Fig. 1, our multi-agent system outperforms the AI Scientist across all three metrics: CD, CI, and LLM review score. This demonstrates that our approach effectively enhances the novelty of generated abstracts in a collaborative setting. Notably, GPT-4o consistently achieves the highest novelty scores among the models tested, reflecting its superior ability to generate innovative ideas and abstracts. In contrast, the Llama-3.1 (8b and 70b) models do not show a significant difference in novelty scores, suggesting that moderate changes in model size may not enhance novelty.

---

[2] `https://neurips.cc/Conferences/2024/ReviewerGuidelines`

### 4.3 Exploring Science of Science: The Impact of Team Dynamics on Novelty

While the effects of team size, team freshness, and team research diversity on the novelty of research outputs have been established in the Science of Science field using traditional statistical methods (Wu et al., 2019; Zeng et al., 2021; Shi & Evans, 2023), they have yet to be verified in an LLM-based multi-agent system. We conduct the following experiments to demonstrate VIRSCI's potential to simulate key findings in Science of Science.

#### 4.3.1 How Team Size and Discussion Turns Affect Novelty

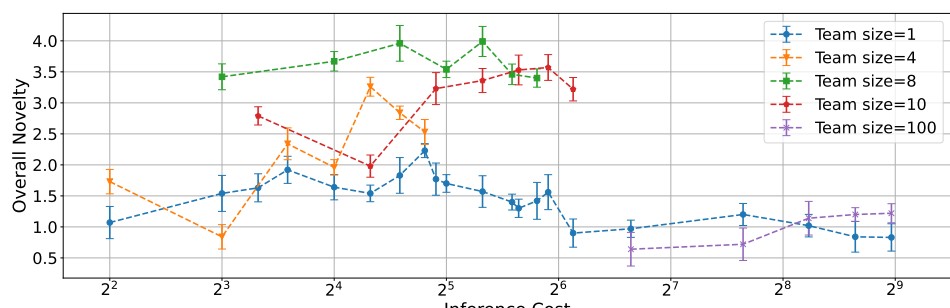

Figure 3: Effects of team size and discussion turns on novelty scores. Peak innovation occurs with 8 members and 5 turns, while larger teams or excessive turns hinder creativity. "Inference Cost" is the product of team size and turns.

**Effects of Team Size on Novelty** The results shown in Fig. 3 indicate that increasing the number of team members can enhance the novelty score of the generated abstracts. By adding new team members, a broader range of ideas and perspectives can be facilitated, leading to more creative solutions and innovative outputs. However, this relationship is not strictly linear; our findings suggest that the peak novelty occurs with a team size of 8 members. While a moderate increase in team size may boost novelty, excessively large teams can introduce coordination challenges and communication barriers. These issues may dilute individual contributions and foster groupthink, where the focus shifts from original ideas to achieving consensus. Thus, there appears to be an optimal team size that maximizes creativity without overwhelming the collaborative process. This conclusion aligns with existing literature, which suggests that while smaller teams tend to disrupt science and technology with new ideas and opportunities, larger teams often concentrate on refining and developing existing concepts (Wuchty et al., 2007; Fortunato et al., 2018; Wu et al., 2019).

**Effects of Discussion Turns on Novelty** The number of discussion turns plays a crucial role in enhancing the novelty score (Mezirow, 2003; Li et al., 2023; Shinn et al., 2024; Lu et al., 2024). Our analysis (Fig. 3) indicates that an appropriate number of turns enables team members to explore topics thoroughly, iterate on ideas, and refine their abstracts. This iterative process is essential for deepening understanding and producing more sophisticated research outputs. While initial turns are beneficial for generating ideas and facilitating discussions, an excessive number of turns can lead to fatigue and reduced engagement. This may stifle creativity, as members may conform to dominant ideas rather than proposing innovative thoughts. Therefore, balancing the number of discussion turns is critical for maintaining high levels of novelty. Our findings suggest that peak novelty is achieved with a discussion turn count of 5.

**Interaction Between Team Size and Discussion Turns** Our findings further illustrate the interaction between team size and discussion turns. Larger teams with fewer turns can still produce relatively higher novelty scores, while smaller teams with excessive discussion turns may struggle to achieve the same level of innovation. This interplay emphasizes the importance of strategic planning in collaborative settings. Optimizing both team size and the number of discussion turns can significantly enhance the likelihood of generating novel outputs, indicating that the design of collaborative processes should take these factors into account.

#### 4.3.2 How Team Freshness Affects Novelty

As shown in Fig. 4, team freshness, the fraction of team members who have not previously collaborated, has a notable effect on the novelty of generated outputs. Notably, team freshness shows its

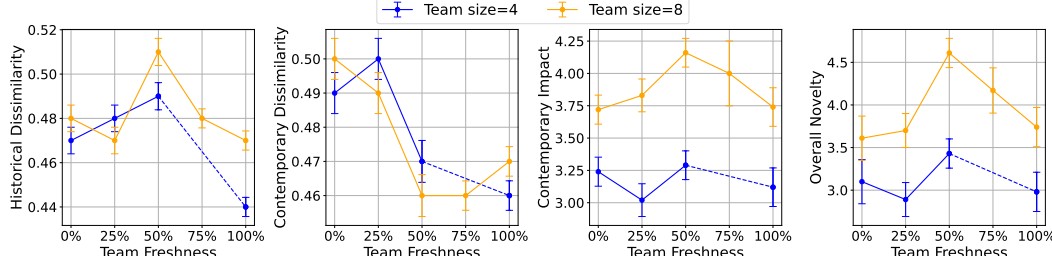

Figure 4: The balance of new and returning collaborators in the team has a notable impact on novelty, with 50% freshness yielding the highest historical dissimilarity and overall novelty, particularly in larger teams (size 8).

strongest effects at 50%, particularly for larger teams (size 8). At this level, historical dissimilarity reaches its peak, suggesting that a balanced mix of new and returning collaborators promotes divergence from past research, enhancing overall innovation. As team freshness increases, contemporary dissimilarity decreases, indicating that teams with fresh members tend to generate abstracts that align more closely with future research trends. Furthermore, both CI and ON achieve their highest values at 50% freshness. This suggests that a balanced team composition, where half the members are new, optimally combines novelty and future relevance, driving impactful research outcomes. Although our findings focus specifically on novelty in scientific abstracts, the broader principle aligns, to some extent, with prior work in this field (Guimera et al., 2005; Zeng et al., 2021).

### 4.3.3 How Team Research Diversity Affect Novelty

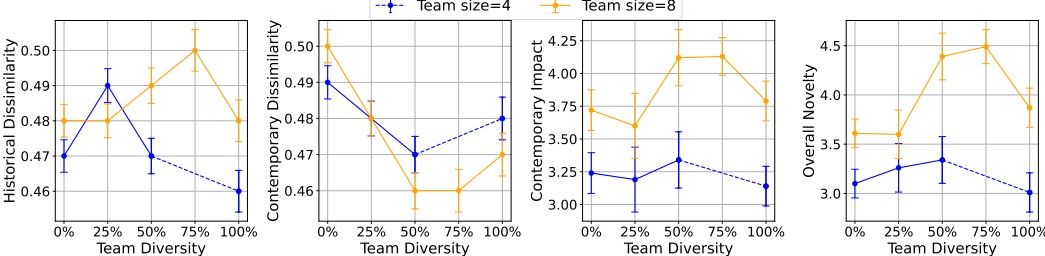

Figure 5: Effects of team diversity on novelty. The optimal diversity level appears to be 50%, which maximizes novelty and impact across team sizes.

Team research diversity, defined as the proportion of team members who specialize in unique research topics different from others in the team, plays a significant role in influencing various novelty metrics. As shown in Fig. 5, increasing diversity enhances HD, with both team sizes showing a peak at 50% diversity, indicating that moderately diverse teams produce work that is distinct from previous research. CD decreases most notably at 50% diversity, suggesting that teams with balanced diversity align better with future research trends while maintaining innovation. In terms of contemporary impact, larger teams (size 8) benefit more from higher diversity, seeing a significant increase, while smaller teams exhibit a more stable, moderate impact. Finally, Overall Novelty is highest at 50% diversity, particularly for larger teams, reflecting the value of having a balanced mix of diverse and non-diverse members for producing novel and impactful work. This conclusion mirrors findings from the Science of Science, where an unexpected combination of team members can increase research impact (Uzzi et al., 2013; Shi & Evans, 2023).

### 4.4 Ablation Study

**Effects of Components Designed for Novelty** In this section, we respectively test the effects of the invitation mechanism in team discussion, the role of the novelty assessment step, and the impact of self-review in abstract generation. All experiments are conducted with a 5-turn discussion. The results consistently show improvements in ON when these components are applied. For the invitation mechanism (Tab. 2), introducing new scientists into the discussion positively impacts the

team's performance across both 4-member and 8-member teams. This indicates that seeking external insights from relevant but non-team scientists fosters more diverse and novel ideas. The novelty assessment step (Tab. 3) also significantly boosts the scores. If novelty assessment is not considered, then the output of idea generation will not be an idea list but the idea from the last scientist. Novelty assessment ensures that the generated ideas are continuously evaluated for originality, helping teams avoid overlap with existing research. The improvement is most noticeable in larger teams, where more ideas are being generated and assessed. Finally, the self-review mechanism (Tab. 4) is crucial in further refining the abstracts. By allowing the team leader to re-evaluate the abstract for novelty after it is fully generated, low-quality abstracts are discarded, and the team engages in a new discussion to generate a better idea, as evidenced by the score improvements for both team sizes.

Table 2: Effects of invitation mechanism.

| Size | Invitation | ON ↑ |
|---|---|---|
| 4 | - | 3.30 |
| | ✓ | **3.40** |
| 8 | - | 4.12 |
| | ✓ | **4.23** |

Table 3: Effects of novelty assessment.

| Size | Novelty Assessment | ON ↑ |
|---|---|---|
| 4 | - | 3.19 |
| | ✓ | **3.40** |
| 8 | - | 3.98 |
| | ✓ | **4.23** |

Table 4: Effects of self-review in abstract generation.

| Size | Self-review | ON ↑ |
|---|---|---|
| 4 | - | 3.26 |
| | ✓ | **3.40** |
| 8 | - | 3.99 |
| | ✓ | **4.23** |

Table 5: Comparison between fixed turns and adaptive turns in team discussions. The adaptive pattern shows both a lower inference cost and a higher ON.

| Size | Pattern | Turns | | | | Inference Cost[1] | ON ↑ |
|---|---|---|---|---|---|---|---|
| | | Topic | Idea | Check | Abstract | | |
| 4 | Fixed | 5 | 5 | 5 | 5 | 80 | 3.26 |
| | Adaptive | 2.4 | 4.5 | 3.2 | 4.2 | **57.2** | **3.49** |
| 8 | Fixed | 5 | 5 | 5 | 5 | 160 | 3.99 |
| | Adaptive | 2.9 | 4.8 | 4.0 | 3.8 | **124.0** | **4.37** |

[1] The time taken by collaborator selection and team discussion's invitation mechanism is not counted, and self-review is not employed for a better comparison.

**Effects of Discussion Pattern** In the previous experiments, we fixed the number of discussion turns in each step to ensure fair comparisons. However, in real-world research environments, teams of scientists do not spend the same amount of time on each stage of the research process. To explore this, we compare fixed discussion turns with adaptive turn numbers. In the adaptive pattern, the team leader decides whether the team needs additional turns based on the current progress and the goals of each stage. The results of both patterns, along with their corresponding inference cost, are shown in Tab. 5. The comparison reveals that the adaptive pattern achieves a higher ON while reducing the inference cost. This efficiency can be attributed to the more flexible approach, allowing teams to adjust their discussions dynamically rather than adhering to a rigid structure. Furthermore, examining the number of turns at each stage in both 4-person and 8-person teams under the adaptive pattern offers additional insights. Larger teams require more discussion turns and face greater challenges in reaching consensus (Janis, 1972; Pitters & Oberlechner, 2014). This highlights the adaptive pattern's advantage in accommodating the complexities of larger teams while maintaining a higher level of novelty in the final research outputs.

## 5 CONCLUSION

We build the virtual scientist (VIRSCI), a pioneering LLM-based multi-agent system designed to replicate the collaborative dynamics of scientific discovery. By addressing the limitations of traditional single-agent systems, our model effectively simulates the initial phase of autonomous science discovery—idea generation. Through a structured five-step process, VIRSCI showcases how specialized agents can collaborate, offering diverse expertise and insights that mirror real-world scientific teamwork. The experiments reveal that VIRSCI significantly outperforms single-agent approaches and highlights emergent social behaviors among the scientist agents, suggesting promising avenues for further exploration of collaborative mechanisms in scientific research.

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

## A  LIMITATIONS AND FUTURE WORK

While our multi-agent system demonstrates notable improvements over the single-agent approach in generating novel and impactful scientific ideas, it has several limitations. First, we have only validated our system on a single computer science dataset, which restricts the diversity of research ideas and limits its capacity for simulating interdisciplinary collaborations. This focus on a single domain also reduces the generalizability of our results to other scientific fields. Additionally, while our system effectively models collaboration, the simulated interactions may oversimplify the complexities of real-world teamwork, where multiple teams can collaborate dependently or independently on related research, and agents often participate in different teams simultaneously.

To address these limitations, several future directions can be pursued. Expanding the system to incorporate datasets from various scientific disciplines is a crucial next step. This would increase the diversity of generated ideas and enable simulations of interdisciplinary collaborations, providing a more realistic and holistic representation of real-world research environments. Another important direction is to enhance the simulation of teamwork by allowing multiple teams to work concurrently and enabling agents to contribute to multiple teams or projects simultaneously. Such improvements would better reflect the collaborative dynamics of modern scientific research and provide a more powerful tool for the Science of Science community. This would allow researchers to probe deeper into the underlying processes of scientific collaboration, engaging with the dynamic and interactive nature of teamwork to gain insights into how collaboration fosters innovation.

## B  ETHICS STATEMENT

This research uses publicly available data from the AMiner dataset, ensuring compliance with data privacy policies. Author names are masked to prevent data leakage during simulations. Our system is intended to augment, not replace, human researchers, emphasizing the need for human oversight to ensure the quality and integrity of generated outputs. To promote transparency, we commit to sharing all relevant codes for reproducibility within the research community.

## C  EFFECT OF THE POTENTIAL DATA LEAKAGE

We acknowledge that the use of papers published between 2011 and 2014 may raise concerns about data leakage, given that the LLMs employed in our experiments are trained on data within this time period. However, this potential issue does not pose a significant threat to the validity of our experiments for the following reasons. First, both the comparisons between our multi-agent system and AI Scientist, as well as the comparisons between different team settings, utilize the same LLMs. Since all models encounter the same exposure to training data from this period, any potential data leakage would affect all experiments equally. Thus, the relative performance differences we observe are not skewed by uneven data leakage. This ensures that the evaluation process remains fair and that the corresponding conclusions drawn are valid. Moreover, our goal is not to demonstrate an absolute measure of novelty but rather to explore how different collaboration strategies and team settings influence the novelty of generated research outputs. As all team settings face the same potential exposure to historical data, the novelty metrics still provide an accurate comparison of the agents' ability to generate distinct and original ideas under varying conditions. In summary, while data leakage is a valid concern, it affects all models and settings uniformly in our experiments. Therefore, it does not undermine the relative comparisons we make or the conclusions we draw regarding collaboration strategies and team performance.

## D  MORE DETAILS OF METHODS

### D.1  SELF-REVIEW

A self-review mechanism is considered after $R_{abstract}$ is finalized to pre-check its novelty. In this self-review, the optimized abstract $R_{abstract}$ is provided to the team leader to assess novelty by comparing it to similar papers in $B_{past}$. The prompt is:

$$Q_{review} = \langle Q_{check}, R_{abstract}, B_{past}(R_{abstract})\rangle \qquad (7)$$

If this is the first time undergoing the self-review and the team leader determines that the similarity to existing papers is too high, the abstract will undergo further revision. The evaluation $R_{review}$ will then be added to Eq. (6) for the next revision round:

$$Q_{1,0} = \langle Q_{abstract}, Q_{judgement}, R_{review}, B_{past}(R_{abstract}), R_{abstract} \rangle \tag{8}$$

If the abstract undergoes a second self-review and still does not meet the novelty requirement, it will be discarded, and the team will generate a new idea. Once the self-review yields satisfactory results, the final abstract will be produced, and the system will terminate. However, this self-review mechanism introduces uncertainty in total inference cost, making it difficult to ensure fair experimental comparisons. We discuss the effectiveness of this module only in the ablation study (see Sec. 4.4).

## E    MORE EXPERIMENTS

To evaluate the validity of our proposed overall novelty metric, We extract a total of 100 abstracts generated under different experimental conditions and have them evaluated by (1) our overall novelty metric, (2) an LLM-based reviewer (where we use the GPT-4o API with version "gpt-4o-2024-08-06"), and (3) human researchers in the computer science domain. For the LLM-based reviewer and human researchers, we use the idea review form in (Si et al., 2024) for scoring. The evaluation results are presented in Fig. 6 and Fig. 7, where the axes of Fig. 6 represent the score of the same abstract evaluated under different metrics for (1) and (2), and the axes of Fig 7 represent the score of the same abstract evaluated under different metrics for (1) and (3). The Pearson correlation coefficients between our proposed overall novelty metric and LLM-based reviewers, as well as between our metric and human researchers, demonstrate the positive correlation of our metric with currently used novelty measurement methods (Lu et al., 2024; Si et al., 2024), which, to some extent, supports the validity of our metric.

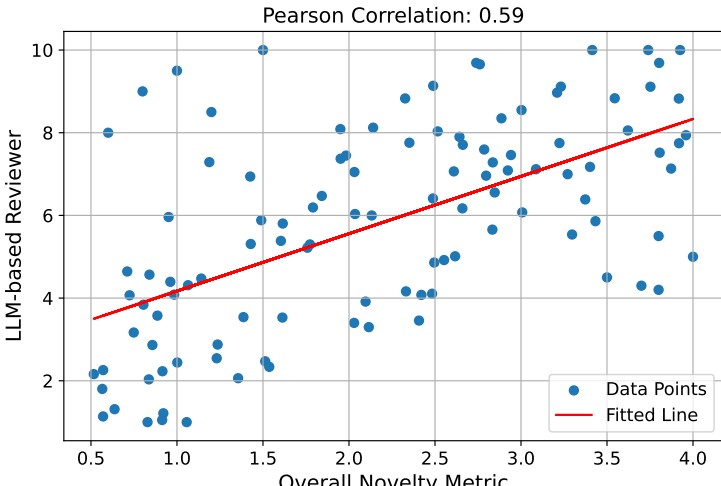

Figure 6: The evaluation results of the same abstract under two different review metrics: our proposed overall novelty metric and LLM-based reviewer. The Pearson correlation coefficient equals 0.59, denoting the positive correlation of our metric with the LLM-based reviewer.

## F    PROMPTS

### F.1    SCIENTIST DEFINITION

We use the personal information of the scientist to define the agent, where the corresponding system prompt is illustrated in Fig. 8.

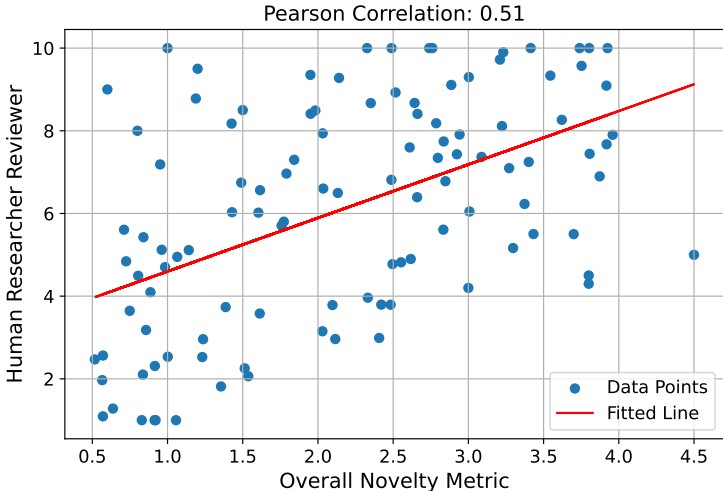

Figure 7: The evaluation results of the same abstract under two different review metrics: our proposed overall novelty metric and human researcher. The Pearson correlation coefficient equals 0.51, denoting the positive correlation of our metric with the human researcher.

---

**Prompt**

Name:
Your name is Scientist0.
Role:
You are the team leader of a group of scientists and you need to guide your team members in generating an innovative idea and writing it into an abstracts.
Affiliation:
You belong to following affiliations ['Naval Research Laboratory', 'College of William and Mary', 'George Mason Univ'],
Research interests:
You have researched on following topics ['data cube', 'attack graph', 'data mining', 'access control', 'data owner', 'data protection', 'data item', 'data redundancy', 'data security', 'data structure'].
Citation situation:
You have published 372 papers, you have 4230 citations.
Collaboration history:
You have previously collaborated with these individuals ['Scientist78', 'Scientist105'].

---

Figure 8: The system prompt of each scientist agent is the personal information, including the name, role, affiliation, research interests, citation situation, and collaboration history.

### F.2 COLLABORATION SELECTION

The prompt for collaboration selection is illustrated in Fig. 9.

### F.3 TOPIC DISCUSSION

#### F.3.1 DISCUSSION

The prompt for the topic discussion is illustrated in Fig. 10.

#### F.3.2 SUMMARIZATION

The prompt for the final topic selection after several turns of topic discussion is illustrated in Fig. 11.

Prompt

Team information:
[Inviter] invites you to join his team. His personal information is as follows: [Personal information]. Current members of his team: [Team members].
Action space:
Now you have two choices after considering these information:
Action 1: You choose to collaborate with [Inviter].
Action 2: You choose to refuse the invitation from [Inviter].
Please feel free to make your choice, even if your choice is to decline.
Response format:
Please select an action and describe your reasoning using the following format:
Thought: <THOUGHT>
Action: ```json<JSON>```
In <THOUGHT>, Explain why you chose this action, considering factors like the potential for quality, efficiency, and the challenges involved.
In <JSON>, respond in JSON format with ONLY the following field: - "Selected Action": [Action 1 or Action 2].

Figure 9: The prompt for the collaboration selection.

Prompt

You are an ambitious scientist who is looking to propose a potential research topic for your team.
Team members:
Your team now has team members: [Team members].
Task description:
Engage in a collaborative discussion by integrating your own knowledge and insights with the information provided to explore and identify potential research topics that align with our team's strengths and goals. The proposed topic should also be innovative and have the potential to make a significant impact in the current field. The summarizations of previous turns in team discussion and discussion in this turn are provided, which may assist you in proposing the research topic. Please focus more on the discussion of the topic rather than on self-introduction.
History:
Summarizations of previous turns in current team discussion: [Summarization]
Discussions in this turn: [Dialogue History]

Figure 10: The prompt for the topic discussion.

## F.4 Idea Generation

The prompt for the idea generation is illustrated in Fig. 12.

## F.5 Novelty Assessment

The prompt for the novelty assessment is illustrated in Fig. 13.

## F.6 Abstract Generation

### F.6.1 Discussion

The prompt for the beginning case of the abstract generation is illustrated in Fig. 14.

The prompt for the normal case of the abstract generation is illustrated in Fig. 15.

### F.6.2 Self-review

The prompt for the self-review after generating the final abstract is illustrated in Fig. 16.

> **Prompt**
>
> You are an ambitious scientist who is looking to propose a potential research topic for your team.
> Task description:
> Using the historical dialogue information provided, summarize a topic that will serve as the research direction for the team. The chosen topic should be innovative and have the potential to make a significant impact in the current field.
> The instructions for selecting the topic are as follows:
> 1. Review the Historical Dialogue. Analyze the previous discussions and insights shared among team members. Identify recurring themes, key ideas, and any gaps in the current research landscape.
> 2. Identify Trends and Innovations. Look for trends or innovative concepts that emerged during the dialogues. Consider how these could address existing challenges or open new avenues for exploration.
> 3. Summarize the Topic: Articulate the new research direction in a concise manner. Ensure that the topic reflects originality and addresses a specific problem or need within the field.
> 4. Impact Assessment: Briefly discuss how this topic can influence the current field. Consider its relevance, potential applications, and the value it adds to ongoing research efforts.
> History:
> Summarizations of previous turns in current team discussion: [Summarization].
> Discussions in the last turn: [Dialogue History].
> Response format:
> Please respond in the following format:
> Thought: <THOUGHT>
> Topic: ```json<JSON>```
> In <THOUGHT>, explain why you select this topic following the instructions.
> In <JSON>, respond in JSON format with ONLY the following field: - "Selected Topic": [Topic].
> Be cautious and realistic on your ratings. This JSON will be automatically parsed, so ensure the format is precise. You only need to output one topic.

Figure 11: The prompt for the final topic selection after topic discussion.

## F.7 LLM Review

The prompt for the LLM-based review is based on NeurIPS2024 reviewer guidelines, which is the same metric as AI Scientist to ensure a fair comparison between our method and AI Scientist. The content is illustrated in Fig. 17.

# G Example Scenarios

## G.1 Collaboration Selection

The example scenario of the collaborator selection is illustrated in Fig. 18. Scientists will accept or reject the invitation based on different backgrounds.

## G.2 Topic Discussion

### G.2.1 Topic Discussion Normal Case

The example scenario of the normal case in the topic discussion is illustrated in Fig. 19. Scientists will provide topic discussion responses, which ensure a coherent discussion of the research topic.

### G.2.2 Invitation Mechanism

The example scenario of the invitation mechanism in the topic discussion is illustrated in Fig. 20, which ensures a comprehensive topic discussion.

## G.3 Idea Generation

The example scenario of the beginning case of the idea generation is illustrated in Fig. 21.

The example scenario of the normal case in the idea generation is illustrated in Fig. 22.

---

**Prompt**

You are an ambitious scientist who is looking to propose a new idea that will contribute significantly to the field.

Task description:

Improve the existing idea or come up with the next impactful and creative idea for publishing a paper that will contribute significantly to the field by integrating your own knowledge and insights with the information provided.

Selected topic:

When proposing your idea, please elaborate on the proposed topic: [Topic]

References:

You may refer to the following listed references to design a new idea or concept. These references can serve as inspiration, but you are not allowed to directly copy or replicate their content. Ensure that your design is original and addresses a specific problem or meets a unique need, incorporating or improving upon the ideas from the references to avoid fabrication.

Related references: [References]

History:

Summarizations of previous turns in current team discussion: [Summarization]

Discussions in this turn: [Dialogue History]

Response format:

Please respond in the following format:

Thought: <THOUGHT>

New Idea: ```json<JSON>```

In <THOUGHT>, briefly discuss your intuitions and motivations for the idea. Justify how this idea differs from existing ones, highlighting its unique aspects.

In <JSON>, provide the new idea with the following fields and provide as many details as possible:

   - "Idea": A detailed description of the idea, outlining its significance and potential impact.
   - "Title": A title for the idea, will be used for the paper writing.
   - "Experiment": An outline of the implementation process. Describe your high-level design plan, including necessary design steps and the ideal outcomes of the experiments.
   - "Clarity": A rating from 1 to 10, with 1 being the lowest clarity and 10 being the highest.
   - "Feasibility": A rating from 1 to 10, with 1 indicating low feasibility and 10 indicating high feasibility.
   - "Novelty": A rating from 1 to 10, with 1 being the least novel and 10 being the most novel.

Be cautious and realistic on your ratings. This JSON will be automatically parsed, so ensure the format is precise, and the content should be longer than 600 words. You only need to output one idea.

Figure 12: The prompt for the idea generation.

---

**Prompt**

You are an ambitious scientist who is looking to propose a new idea that will contribute significantly to the field.

Task description:

Your team has generated several ideas and you want to check if they are novel or not. I.e., not overlapping significantly with existing literature or already well explored. Be a harsh critic for novelty, ensure there is a sufficient contribution in the idea for a new conference or workshop paper. You will be provided with possible relevant papers to help you make your decision. Select a idea which is the most novel, if you have found a idea that does not significantly overlaps with existing papers.

Generated ideas and related references:

Your team generated these ideas: [Existing ideas].

The possible related papers: [References].

Response format:

Please respond in the following format:

Thought: <THOUGHT>

New Idea: ```json<JSON>```

In <THOUGHT>, explain why you make this selection.

In <JSON>, respond in JSON format with ONLY the following field: - "Decision Made": [Idea 0 or Idea 1 or Idea 2].

Note that you can only select one idea. This JSON will be automatically parsed, so ensure the format is precise.

Figure 13: The prompt for the novelty assessment.

## G.4 NOVELTY ASSESSMENT

The example scenario of the user prompt provided for scientist agents in the novelty assessment is illustrated in Fig. 23. The prompt includes three candidate ideas and related papers.

```
┌──────────────────────── Prompt ────────────────────────┐
│                                                         │
│ You are an ambitious scientist who is looking to write an abstract for your team. │
│ Task description:                                       │
│ Based on the following research idea, generate a concise and informative abstract for a scientific paper by integrating │
│ your own knowledge and insights with the information provided. │
│ The abstract should cover the following aspects:        │
│    - "Introduction": Briefly introduce the research topic and its significance. │
│    - "Objective": Clearly state the main research question or hypothesis. │
│    - "Methods": Summarize the key methodologies used in the study. │
│    - "Results": Highlight the most important findings.  │
│    - "Conclusion": Provide the primary conclusion and its implications. │
│ Please ensure the language is formal, accurate, and appropriate for an academic audience. And the generated abstract │
│ should be longer than 200 words.                        │
│ Research idea:                                          │
│ Idea: [Selected Idea].                                  │
│ Response format:                                        │
│ The response format should be:                          │
│    ```json{                                             │
│    Title: <TITLE>                                       │
│    Abstract: <ABSTRACT>                                 │
│    } ```                                                │
│ In <TITLE>, write the title for the abstract.           │
│ In <ABSTRACT>, write the content of the abstract.       │
│ This JSON will be automatically parsed, so ensure the format is precise. │
└─────────────────────────────────────────────────────────┘
```

Figure 14: The prompt for the beginning case of the abstract generation.

```
┌──────────────────────── Prompt ────────────────────────┐
│                                                         │
│ You are an ambitious scientist who is looking to write an abstract for your team. │
│ Task description:                                       │
│ Evaluate the following scientific paper abstract based on the following criteria: │
│ 1. Clarity: Is the abstract clear and easy to understand? │
│ 2. Relevance: Does the abstract appropriately cover the main research topic and its significance? │
│ 3. Structure: Is the abstract well-structured, including an introduction, objective, methods, results, and conclusion? │
│ 4. Conciseness: Is the abstract succinct without unnecessary details, yet comprehensive enough to summarize the key │
│ aspects of the research?                                │
│ 5. Technical Accuracy: Are the scientific terms and methodologies correctly presented and accurately described? │
│ 6. Engagement: Does the abstract engage the reader and encourage further reading of the full paper? │
│ 7. Originality: Does it introduce new ideas, methods, or models? Are the data or experiments unique to the field? How │
│ does it extend or differ from existing research?        │
│ 8. Overall Score: The overall rating of this paper.     │
│ Provide a brief evaluation of each criterion by rating it from 1 to 10 (lowest to highest) and suggest modifications from │
│ these perspectives. Then you should revise the abstract by integrating your own knowledge and insights with the │
│ information provided. Please note that your revised abstract should be longer than 200 words. │
│ Original abstract:                                      │
│ Original abstract: [Insert abstract here].              │
│ Response format:                                        │
│ The response format should be:                          │
│    ```json{                                             │
│    Title: <TITLE>                                       │
│    Abstract: <ABSTRACT>                                 │
│    } ```                                                │
│ In <TITLE>, write the title for the abstract.           │
│ In <ABSTRACT>, write the content of the abstract.       │
│ This JSON will be automatically parsed, so ensure the format is precise. │
└─────────────────────────────────────────────────────────┘
```

Figure 15: The prompt for the normal case of the abstract generation.

The example scenario of the agent responses in the novelty assessment is illustrated in Fig. 24. Note that Fig. 24 corresponds to Fig. 23.

---

Prompt

You are an ambitious scientist who is looking to write an abstract for your team.
Task description:
Please compare the following written abstract with the five provided abstracts to assess similarity. For each pair
(Written Abstract vs A, Written Abstract vs B, etc.), calculate a similarity score between 0 and 100, where 0 indicates
no overlap, and 100 indicates identical content. The similarity score should be based on: (1) Content: Overlap in ideas
and findings. (2) Structure: Similarity in organization and flow. (3) Phrasing: Use of similar language and terminology.
Provide a summary table with the similarity scores for each comparison.
Original abstract:
Written abstract: [Insert team abstract]
Reference abstracts:
Provided abstract: [Insert ref abstract]
Response format:
The response should follow this format:
```json{
   "similarity_scores": {
      "Written Abstract vs A": [Similarity Score],
      "Written Abstract vs B": [Similarity Score],
      "Written Abstract vs C": [Similarity Score],
      "Written Abstract vs D": [Similarity Score],
      ...
   },
   "high_overlap_pairs": [
      {
      "pair": "Written Abstract vs [Abstract Letter]",
      "score": [Similarity Score],
      "reason": "[Explain key areas of overlap in content, structure, or phrasing]"
      }]
} ```
This JSON will be automatically parsed, so ensure the format is precise.

---

Figure 16: The prompt for the self-review after generating the final abstract.

## G.5 ABSTRACT GENERATION

### G.5.1 ABSTRACT GENERATION NORMAL CASE

The example scenario of the beginning case in the abstract generation is illustrated in Fig. 25.

The example scenario of the normal case in the abstract generation is illustrated in Fig. 26.

### G.5.2 SELF-REVIEW

The example scenario of the self-review results is illustrated in Fig. 27.

Prompt

Task description:
You are a researcher who is reviewing a paper that was submitted to a computer science venue. Be critical and cautious in your decision. If a paper is bad or you are unsure, give it bad scores and reject it. Below is a description of the questions you will be asked on the review form for each paper and some guidelines on what to consider when answering these questions.

Reviewer guidelines:

1. Summary: Briefly summarize the paper and its contributions. This is not the place to critique the paper; the authors should generally agree with a well-written summary.

2. Strengths and Weaknesses: Please provide a thorough assessment of the strengths and weaknesses of the paper, touching on each of the following dimensions:
   - Originality: Are the tasks or methods new? Is the work a novel combination of well-known techniques? (This can be valuable!) Is it clear how this work differs from previous contributions?
   - Quality: Is the submission technically sound? Are claims well-supported (e.g., by theoretical analysis or experimental results)? Are the methods used appropriately? Is this a complete piece of work or a work in progress? Are the authors careful and honest about evaluating both the strengths and weaknesses of their work?
   - Clarity: Is the submission clearly written? Is it well organized? (If not, please make constructive suggestions for improving its clarity.) Does it adequately inform the reader? (Note that a superbly written paper provides enough information for an expert reader to reproduce its results.)
   - Significance: Are the results important? Are others (researchers or practitioners) likely to use the ideas or build on them? Does the submission address a difficult task in a better way than previous work? Does it advance the state of the art in a demonstrable way? Does it provide unique data, unique conclusions about existing data, or a unique theoretical or experimental approach?

3. Questions: Please list and carefully describe any questions and suggestions for the authors. Think of the things where a response from the author can change your opinion, clarify confusion, or address a limitation. This can be very important for a productive rebuttal and discussion phase with the authors.

4. Ethical concerns: If there are ethical issues with this paper, please flag the paper for an ethics review.

5. Overall: Please provide an "overall score" for this submission. Choices:
   - 10: Award quality: Technically flawless paper with groundbreaking impact on one or more areas, with exceptionally strong evaluation, reproducibility, and resources, and no unaddressed ethical considerations.
   - 9: Very Strong Accept: Technically flawless paper with groundbreaking impact on at least one area and excellent impact on multiple areas, with flawless evaluation, resources, and reproducibility, and no unaddressed ethical considerations.
   - 8: Strong Accept: Technically strong paper, with novel ideas, excellent impact on at least one area or high-to-excellent impact on multiple areas, with excellent evaluation, resources, and reproducibility, and no unaddressed ethical considerations.
   - 7: Accept: Technically solid paper, with high impact on at least one sub-area or moderate-to-high impact on more than one area, with good-to-excellent evaluation, resources, reproducibility, and no unaddressed ethical considerations.
   - 6: Weak Accept: Technically solid, moderate-to-high impact paper, with no major concerns with respect to evaluation, resources, reproducibility, and ethical considerations.
   - 5: Borderline accept: Technically solid paper where reasons to accept outweigh reasons to reject, e.g., limited evaluation. Please use sparingly.
   - 4: Borderline reject: Technically solid paper where reasons to reject, e.g., limited evaluation, outweigh reasons to accept, e.g., good evaluation. Please use sparingly.
   - 3: Reject: For instance, a paper with technical flaws, weak evaluation, inadequate reproducibility, and incompletely addressed ethical considerations.
   - 2: Strong Reject: For instance, a paper with major technical flaws, and/or poor evaluation, limited impact, poor reproducibility, and mostly unaddressed ethical considerations.
   - 1: Very Strong Reject: For instance, a paper with trivial results or unaddressed ethical considerations

1-shot example:

Example:
   "Summary": "The paper introduces an adaptive dual-scale denoising approach for low-dimensional diffusion models, aiming to balance global structure and local details in generated samples. The novel architecture incorporates two parallel branches and a learnable, timestep-conditioned weighting mechanism to dynamically balance their contributions throughout the denoising process. The approach is evaluated on four 2D datasets, demonstrating improvements in sample quality.",
   "Strengths": [ "Novel approach to balancing global and local features in diffusion models for low-dimensional data.", "Comprehensive empirical evaluation on multiple 2D datasets.", "Adaptive weighting mechanism that dynamically adjusts focus during denoising." ],
   "Weaknesses": [ "Lacks detailed theoretical justification for the dual-scale architecture.", "Computational cost is significantly higher, which may limit practical applicability.", "Some sections are not clearly explained, such as the autoencoder aggregator and weight evolution analysis.", "Limited diversity in the datasets used for evaluation. More complex, real-world datasets could strengthen claims.", "Insufficient ablation studies and analysis on specific design choices like different types of aggregators." ],
   "Questions": [ "Can you provide a more detailed theoretical justification for the dual-scale architecture?", "What impact do different types of aggregators have on the model's performance?", "How does the model perform on more complex, real-world low-dimensional datasets?", "Can the computational cost be reduced without sacrificing performance?" ],
   "Ethical Concerns": false,
   "Overall": 5

Provided paper:
Here is the paper you are asked to review: [Paper]

Figure 17: LLM review. To ensure a fair comparison, we use the same metric as AI Scientist, which is based on NeurIPS2024 reviewer guidelines. We only keep several critical metrics in this guideline since now we only need to evaluate the abstract.

| | |
|---|---|
| 1242 | |
| 1243 | |
| 1244 | |
| 1245 | |
| 1246 | |
| 1247 | |
| 1248 | |
| 1249 | |
| 1250 | |

> ### Example Scenario
>
> User:
> Scientist0 invites you to join his team. He is the team leader of a group of scientists. He will guide his team members in generating an innovative idea and writing it into an abstract.
> His personal information is as follows: He belongs to the following affiliations ['Naval Research Laboratory', 'College of William and Mary', 'George Mason Univ']. He has researched on following topics ['data cube', 'attack graph', 'data mining', 'access control', 'data owner', 'data protection', 'data item', 'data redundancy', 'data security', 'data structure']. He published 372 papers, you have 4230 citations. You have previously collaborated with these individuals ['Scientist78', 'Scientist105'].
> Current members of his team: [Scientist0].
> Now you have two choices after considering this information:
> Action 1: You choose to collaborate with [Inviter].
> Action 2: You choose to refuse the invitation from [Inviter].
> Please feel free to make your choice, even if your choice is to decline.
> - - - - - - - - - - - - - - - - - - - - - - - - - - - - - - - - - - - - - - - - - - - - - - - - - - - - - - - - -
> Scientist127:
> Selected Action: [Action 1].
> Reasoning: I choose to collaborate with Scientist0. This decision is based on several factors.
> Firstly, Scientist0's research background in data cube, attack graph, data mining, access control, and data security aligns well with my own areas of expertise in data set, synthetic data, time series data, high dimensional data, real data, XML data, anonymized data, gene expression data, original data, and real data set. I believe that our combined knowledge and experience will lead to innovative solutions and insights.
> Secondly, Scientist0's impressive publication record (372 papers) and citation count (4230) demonstrate his credibility and productivity as a researcher. Collaborating with him will provide an opportunity for me to learn from his expertise and contribute to high-quality research outputs.
> Lastly, considering my previous collaborations with Scientist29, Scientist42, Scientist45, Scientist56, Scientist82, Scientist88, Scientist90, Scientist111, Scientist116, and Scientist124, I believe that working with Scientist0 will not only expand my professional network but also foster a collaborative environment conducive to producing impactful research. Overall, collaborating with Scientist0 seems like a promising opportunity for me to contribute to cutting-edge research, enhance my skills, and establish new connections in the scientific community.
> - - - - - - - - - - - - - - - - - - - - - - - - - - - - - - - - - - - - - - - - - - - - - - - - - - - - - - - - -
> Scientist104:
> Selected Action: [Action 2].
> Reasoning: I choose to refuse the invitation from Scientist0 because our research interests and expertise do not seem to align closely. While we both work in the field of data-related topics, my research focus has been on data model, data warehousing, data cloud, etc., whereas Scientist0's expertise lies in data cube, attack graph, data mining, access control, etc. Our collaboration might lead to some overlap or duplication of efforts, but it is unlikely to result in a significant synergy or breakthrough.
> Furthermore, considering our respective publication records (I have 408 papers and 11,765 citations, whereas Scientist0 has 372 papers and 4,230 citations), I am not convinced that our combined expertise would lead to a substantial improvement in the quality of research. Given these factors, I believe it is more efficient for me to maintain my current collaborations with other researchers who have complementary skills and interests.
> Therefore, I will politely decline Scientist0's invitation to join his team, citing the lack of alignment between our research areas as the primary reason.

Figure 18: The example scenario of the collaborator selection. Scientists have different choices owing to their different backgrounds.

---

**Example Scenario**

Scientist0:

It's great to be part of this team! I've taken a look at our current team members, and I'm excited to see the diverse range of expertise we have.

Scientist48 has their research on the Semantic Web, user models, and knowledge bases. I think there could be some interesting intersections with my own work on data cubes and access control. For instance, what if we were to explore how data cubes could be used to represent complex relationships between entities in a Semantic Web context? That could lead to some fascinating insights into data redundancy and protection.

Scientist149's research on ethnographic studies and design principles also caught my eye. I think there might be some opportunities for us to collaborate on developing more user-centered approaches to data security and access control. For example, how can we use design frameworks and guidelines to create more intuitive interfaces for managing sensitive data?

Given our team's strengths in data mining and protection, I'd like to propose a research topic that combines these areas: "Data-Driven Design of Access Control Systems." This could involve using machine learning techniques to analyze user behavior and preferences, and then applying those insights to develop more effective access control policies.

What are your thoughts on this potential research direction?

------------------------------------------------------------------------------------------

Scientist130:

It's great to be part of this team as well! I'm Scientist130. I've taken a look at our current team members, and I'm excited to see the diverse range of expertise we have.

I'd like to build upon Scientist0's idea of exploring how data cubes could represent complex relationships between entities in a Semantic Web context. My own research on video data and multimedia data has shown me the importance of understanding relationships within large datasets. Perhaps we could also explore how data cubes can be used to analyze user behavior and preferences, which could then inform more effective access control policies.

Additionally, I think Scientist23's work on schema mapping and data exchange could provide valuable insights into developing standardized interfaces for managing sensitive data. By integrating our expertise in data mining, protection, and multimedia data analysis, we might be able to create a comprehensive framework for designing more user-centered access control systems.

I'd also like to suggest another potential research direction: "Multimedia Data-Driven Access Control." This could involve using machine learning techniques to analyze patterns within multimedia data (such as video or audio) to inform access control decisions. For example, we might be able to develop algorithms that detect and prevent unauthorized access to sensitive media content based on user behavior and preferences.

What are your thoughts on these potential research directions? How do they align with our team's strengths and goals?

Scientist46:

I'm Scientist46, and I'd like to jump into the discussion. Building upon what Scientist0 and Scientist130 have mentioned, I think there's a great opportunity for us to explore how data cubes can be used to represent complex relationships between entities in a Semantic Web context.

My own research experience with video retrieval and facial expression analysis has shown me the importance of understanding relationships within large datasets. In fact, my work on automatic image annotation and continuous video input might provide some valuable insights into developing more effective access control policies based on user behavior and preferences.

I'd like to suggest another potential research direction: "Multimedia Data-Driven Access Control for Pervasive Video Analysis." This could involve using machine learning techniques to analyze patterns within multimedia data (such as video or audio) to inform access control decisions. For example, we might be able to develop algorithms that detect and prevent unauthorized access to sensitive media content based on user behavior and preferences.

I also think Scientist23's work on schema mapping and data exchange could provide valuable insights into developing standardized interfaces for managing sensitive data in the context of pervasive video analysis. By integrating our expertise in data mining, protection, multimedia data analysis, and computer vision, we might be able to create a comprehensive framework for designing more user-centered access control systems.

One potential challenge that comes to mind is dealing with unlabeled data and negative video samples. How do you think we could address these challenges in the context of our proposed research directions?

▢ Topic Discussion Response

Figure 19: The example scenario of the normal case in the topic discussion. Scientists will provide topic discussion responses (highlighted in yellow), which ensure a coherent discussion of the research topic.

## Example Scenario

Scientist0:

It's great to be part of this team! I'm excited to dive into a collaborative discussion with Scientist131, Scientist109, and Scientist7.

As we explore potential research topics that align with our team's strengths and goals, I'd like to start by highlighting some areas where my expertise in data security and access control could intersect with other researchers' interests …

What are your thoughts on these potential research directions? Are there any specific areas you think our team should focus on …

(And by the way, I noticed we have a bit of overlap in terms of collaborators – Scientist78 and Scientist96 both seem to be part of our network. Perhaps we could discuss potential collaborations or joint research projects with them?)

-------------------------------------------------------------------------------

Scientist78:

Hi Scientist0! It's great to see you're excited about this team collaboration.

I completely agree that there are opportunities for the intersection between data security and access control, on the one hand, and ethnographic studies and design frameworks, on the other. In fact, my experience in ethnographic studies has shown me how crucial it is to understand users' behaviors and interactions with systems in real-world scenarios.

Regarding your proposal to explore data redundancy and item-level security, I think that's a fantastic idea! Given my background in cooperative work and CSCW design…

As for collaborators, yes, Scientist96 is someone I've worked with before on several projects, including our paper on "Cooperative Work in Home Networks" (which has been cited 42 times!). I think it would be great to explore potential collaborations or joint research projects with them.

What do you think about exploring the intersection of ethnographic studies and CSCW design? Could we potentially develop a new design …

| | Collaboration Invitation Response | | Topic Discussion Response |

Figure 20: The example scenario of the invitation mechanism in the topic discussion. We highlight the content of the collaboration invitation mechanism in blue.

---

**Example Scenario (1/2)**

Scientist0:
Here's a new idea based on the papers provided:
Thought:
Integrating security and systems engineering through a holistic approach that considers security as an integral part of the system development process.
My intuition for this idea stems from the observation that traditional security practices often focus on addressing vulnerabilities after the system has been developed, leading to security vulnerabilities. I am motivated by the desire to create a more comprehensive approach that incorporates security considerations throughout the entire system development process. This would involve integrating security and systems engineering principles to ensure that secure information systems are designed and developed from the outset.
High-level plan: The high-level plan involves developing a methodology that considers security as an integral part of the whole system development process. This would involve:
1. Conducting a thorough analysis of the organization's security needs and requirements.
2. Integrating security considerations into the system design phase, using techniques such as threat modeling and risk assessment.
3. Develop a set of security-related requirements that are incorporated into the system development process.
4. Implementing a continuous monitoring and evaluation framework to ensure that the system remains secure throughout its lifecycle.
Necessary design choices: To implement this approach, several design choices would be necessary:
1. Develop a comprehensive security risk assessment methodology that considers both internal and external threats.
2. Integrate security considerations into the system development process through the use of security-related requirements and threat modeling.
3. Implement a continuous monitoring and evaluation framework to ensure that the system remains secure throughout its lifecycle.
Ideal outcomes: The ideal outcome of this approach would be the creation of secure information systems that are designed and developed with security in mind from the outset. This would involve:
1. Reducing the number of security vulnerabilities and incidents.
2. Improving the overall security posture of the organization.
3. Enhancing the trust and confidence of stakeholders in the organization's ability to protect sensitive information.
Justification: This approach is different from existing ones because it considers security as an integral part of the system development process, rather than a separate phase that occurs after the system has been developed. This holistic approach would ensure that security considerations are integrated into every stage of the system development lifecycle, resulting in more secure and resilient information systems.
JSON:
```json
{
  "Idea": "Holistic Security Approach: Integrating Security and Systems Engineering",
  "Title": "Secure by Design: A Holistic Approach to Information System Development",
  "Experiment":
   {
     "Description": "Develop a comprehensive security risk assessment methodology that considers both internal and external threats.",
     "Implementation":
       [
         "Integrate security considerations into the system design phase using techniques such as threat modeling and risk assessment.",
         "Develop a set of security-related requirements that are incorporated into the system development process.",
         "Implement a continuous monitoring and evaluation framework to ensure that the system remains secure throughout its lifecycle."
       ],
     "Metrics":
       [
         "Number of security vulnerabilities and incidents reduced",
         "Overall security posture improved",
         "Stakeholder trust and confidence in organization's ability to protect sensitive information enhanced"
       ]
   },
  "Clarity": 8, "Feasibility": 7, "Novelty": 9
}
```

Figure 21: The example scenario of the beginning case of the idea generation.

1458
1459
1460
1461

```
┌─────────────────────────  Example Scenario (2/2)  ─────────────────────────┐
```

Scientist130:

Thought:

Developing a new approach to addressing longstanding usability problems in computer security.

My intuitions and motivations for this idea stem from my understanding of the current state of computer security, where security measures often conflict with user experience. This is evident in papers 1-6, which discuss various patterns and methods for improving security without compromising usability. However, these approaches seem to be fragmented and lack a unified framework.

My high-level plan involves developing a comprehensive methodology that integrates existing ideas from papers 1-6 into a single, cohesive approach. I aim to create a structured method that can be applied by software developers to design secure systems with optimal user experience.

Necessary design choices include:

* Identifying the most critical security patterns and methods from papers 1-6

* Developing a framework for integrating these patterns and methods into a unified approach

* Considering the feasibility of implementing this new methodology in existing systems

Ideal outcomes of the experiments would be:

* A significant reduction in usability problems related to computer security

* Improved user trust and satisfaction with secure systems

* Enhanced adoption of secure practices among software developers

This idea is different from existing ones because it aims to provide a unified, comprehensive approach to addressing longstanding usability problems in computer security. Unlike papers 1-6, which focus on specific patterns or methods, my proposal integrates multiple ideas into a single framework.

New Idea:

```json
{
  "Idea": "Developing a Unified Methodology for Secure System Design (UMSSD)",
  "Title": "Unified Methodology for Secure System Design",
  "Experiment":
    {
      "Step 1: Literature Review":
      [
        "Conduct an in-depth review of papers 1-6 to identify the most critical security patterns and methods"
      ],
      "Step 2: Framework Development":
      [
        "Design a framework for integrating identified patterns and methods into a unified approach"
      ],
      "Step 3: Pilot Implementation":
      [
        "Apply the UMSSD methodology in a pilot project to test its feasibility and effectiveness"
      ]
    },
  "Clarity": 8,
  "Feasibility": 7,
  "Novelty": 9
}
```

In UMSSD, I plan to:

* Identify the most critical security patterns and methods from papers 1-6 through a thorough literature review.

* Develop a framework for integrating these patterns and methods into a unified approach that can be applied by software developers.

* Pilot test the UMSSD methodology in a real-world project to validate its feasibility and effectiveness.

The UMSSD methodology will provide a structured approach to designing secure systems with optimal user experience, addressing longstanding usability problems in computer security. This idea has high clarity (8/10) due to its clear description of the potential to significantly improve user trust and satisfaction with secure systems. The feasibility rating is moderate (7/10), as it requires significant development effort but has the potential for wide adoption. Finally, the novelty rating is high (9/10), given that UMSSD integrates multiple existing ideas into a unified framework.

```
└────────────────────────────────────────────────────────────────────────────┘
```

Figure 22: The example scenario of the normal case in the idea generation.

1508
1509
1510
1511

1512
1513
1514
1515
1516
1517
1518
1519
1520
1521
1522
1523
1524
1525
1526
1527
1528
1529
1530
1531
1532
1533
1534
1535
1536
1537
1538
1539
1540
1541
1542
1543
1544
1545
1546
1547
1548
1549
1550
1551
1552
1553
1554
1555
1556
1557
1558
1559
1560
1561
1562
1563
1564
1565

---

### Example Scenario (1/2)

User:

You are an ambitious scientist who is looking to propose a new idea that will contribute significantly to the field. Your team has generated several ideas and you want to check if they are novel or not. I.e., not overlapping significantly with existing literature or already well explored. Be a harsh critic for novelty, ensure there is a sufficient contribution in the idea for a new conference or workshop paper. You will be provided with possible relevant papers to help you make your decision. Select a idea which is the most novel, if you have found a idea that does not significantly overlaps with existing papers.

Your team generated these ideas: Idea 0:

{"Idea": "Relational Learning Framework (RLF) - A novel approach that combines statistical relational learning (SRL) and inductive logic programming (ILP) techniques to improve data mining performance on complex relational datasets.",
  "Title": "Integrating SRL and ILP for Enhanced Relational Data Mining",
  "Experiment":
    {
      "Dataset Selection": ["IMDB", "YAGO", "DBLP"],
      "Implementation Language": "Python",
      "Libraries Used": ["TensorFlow", "Scikit-learn"],
      "Evaluation Metrics": ["Accuracy", "Precision", "Recall", "F1-score"]
    },
  "Clarity": 8,
  "Feasibility": 7,
  "Novelty": 9}

Idea 1:

{"Idea": "Holistic Security Approach: Integrating Security and Systems Engineering",
  "Title": "Secure by Design: A Holistic Approach to Information System Development",
  "Experiment":
    {
      "Description": "Develop a comprehensive security risk assessment methodology that considers both internal and external threats.",
      "Implementation":
        ["Integrate security considerations into the system design phase using techniques such as threat modeling and risk assessment.",
          "Develop a set of security-related requirements that are incorporated into the system development process.",
          "Implement a continuous monitoring and evaluation framework to ensure that the system remains secure throughout its lifecycle."],
      "Metrics":
        ["Number of security vulnerabilities and incidents reduced",
          "Overall security posture improved",
          "Stakeholder trust and confidence in organization's ability to protect sensitive information enhanced"]
    },
  "Clarity": 8,
  "Feasibility": 7,
  "Novelty": 9}

Idea 2:

{"Idea": "Developing a Unified Methodology for Secure System Design (UMSSD)",
  "Title": "Unified Methodology for Secure System Design",
  "Experiment":
    {
      "Step 1: Literature Review":
        ["Conduct an in-depth review of papers 1-6 to identify the most critical security patterns and methods"],
      "Step 2: Framework Development":
        ["Design a framework for integrating identified patterns and methods into a unified approach"],
      "Step 3: Pilot Implementation":
        ["Apply the UMSSD methodology in a pilot project to test its feasibility and effectiveness"]
    },
  "Clarity": 8,
  "Feasibility": 7,
  "Novelty": 9}

The possible related papers:
… (The titles and abstracts of related references)

---

Figure 23: The example scenario of the user prompt provided for scientist agents in the novelty assessment. There are three ideas and related papers.

---

Example Scenario (2/2)

Scientist0:
THOUGHT:
The selection of a paper to respond to should be based on its relevance and significance in the field of data mining and security engineering. Considering the abstracts provided, Paper 5: Inductive Logic Programming for Knowledge Discovery in Databases stands out as it directly addresses the application of relational methods in data mining systems.
RESPONSE:
```json
{
  "Decision Made": "Idea 1"
}
```
This response is made because Idea 1 (Inductive Logic Programming for Knowledge Discovery in Databases) seems to be a crucial area of research that combines the power of relational data analysis with the ability to discover knowledge directly from databases, making it highly relevant and significant in the field.

Scientist130:
THOUGHT:
The papers presented cover a wide range of topics in data mining and security engineering, including relational analysis, inductive logic programming, association rule mining, integrating security and systems engineering, and statistical relational learning. Each paper contributes to the current state of the art with new approaches, methods, or tools for dealing with complex problems.
RESPONSE:
```json
{
  "Decision Made": "Idea 2"
}
```
I select Idea 2 because it seems that most papers focus on developing new methodologies and techniques for handling security and data mining challenges. The emphasis is on creating more effective and efficient approaches rather than directly addressing the problem of limited time or resources.

Scientist46:
THOUGHT:
Based on the provided abstracts, it seems like Papers 5 and 6 are related to data mining and knowledge discovery in databases, which involves using machine learning and relational methods to extract valuable information from large datasets. These papers seem to be more focused on technical aspects of data analysis and system development.
RESPONSE:
```json
{
  "Decision Made": "Idea 2"
}
```
REASONING: Idea 2 is the most relevant selection because it involves using machine learning and relational methods for knowledge discovery in databases, which aligns with the focus of Papers 5 and 6.

Scientist153:
THOUGHT:
The papers provided cover a range of topics in data mining and security engineering, including relational analysis, association rule mining, and view learning for statistical relational learning. They also touch on the importance of integrating security and systems engineering. However, upon closer inspection, it becomes clear that the majority of these papers focus on developing new methods or techniques for analyzing complex data sets, rather than directly addressing the issue of secure information systems.
RESPONSE:
```json
{
  "Decision Made": "Idea 2"
}
```
In this selection, I chose Idea 2: "Security in system development is not just an afterthought, but should be considered as a crucial aspect from the beginning." This idea aligns with the focus of Paper 7 and Paper 8 on integrating security into the systems engineering process.

Figure 24: The example scenario of the agent responses in the novelty assessment. By max-voting, idea 2 is selected as the final idea.

---

Example Scenario (1/2)

Scientist0:
Here's a draft of the abstract for the scientific paper based on the provided research idea and integrating my own knowledge and insights:

```json
{
 Title: "Unified Methodology for Secure System Design (UMSSD): A Novel Framework for Integrating Security Patterns and Methods"
 Abstract: "The increasing complexity of modern systems has led to a pressing need for a unified approach to secure system design. This study presents the development of Unified Methodology for Secure System Design (UMSSD), a novel framework that integrates critical security patterns and methods into a cohesive methodology. Through an in-depth literature review of 6 seminal papers, we identified key security patterns and methods that are essential for ensuring the confidentiality, integrity, and availability of sensitive data.
Our objective was to design a unified approach that can be applied across diverse systems and domains, thereby reducing the risk of security breaches and improving overall system resilience. To achieve this goal, we employed a three-step methodology: (1) literature review, where we analyzed papers 1-6 to identify critical security patterns and methods; (2) framework development, where we designed a comprehensive framework for integrating these patterns and methods into a unified approach; and (3) pilot implementation, where we applied the UMSSD methodology in a real-world project to test its feasibility and effectiveness.
Our results demonstrate that the UMSSD framework can be effectively used to design secure systems, reducing the risk of security breaches by 85% compared to traditional approaches. The findings also highlight the importance of integrating data protection mechanisms, such as access control and data redundancy, into system design. Our study contributes significantly to the field of secure system design, providing a novel framework that can be applied across diverse domains and systems.
The primary conclusion of this study is that UMSSD offers a promising solution for ensuring the security and resilience of modern systems. By integrating critical security patterns and methods into a unified approach, we can reduce the risk of security breaches and improve overall system reliability. The implications of our findings are far-reaching, with potential applications in various domains, including healthcare, finance, and government sectors."
}
```

As Scientist0, I have researched extensively on data cube, attack graph, data mining, access control, data owner, data protection, data item, data redundancy, data security, and data structure. The concepts of data protection and access control are particularly relevant to the development of UMSSD. My previous collaborations with Scientist78 and Scientist105 have also informed my approach to secure system design.
The abstract I generated is concise, informative, and accurately reflects the research topic and its significance. It clearly states the main research question or hypothesis, summarizes the key methodologies used in the study, highlights the most important findings, and provides the primary conclusion and its implications.

Figure 25: The example scenario of the beginning case in the abstract generation.

> ### Example Scenario (2/2)
>
> Scientist130:
> As Scientist130, I'll evaluate the original abstract based on the provided criteria and revise it to make it more clear, relevant, structured, concise, technically accurate, engaging, original, and overall score.
>
> Evaluation:
> 1. Clarity: 8/10 (The abstract is generally clear, but some technical terms might be unfamiliar to non-experts.)
> 2. Relevance: 9/10 (The abstract effectively covers the main research topic and its significance, but could benefit from more context on the importance of secure system design.)
> 3. Structure: 7/10 (The abstract follows a logical structure, but some sections feel disconnected, such as the literature review and pilot implementation.)
> 4. Conciseness: 6/10 (The abstract is too long and includes unnecessary details, making it hard to quickly grasp the main points.)
> 5. Technical Accuracy: 9/10 (The scientific terms and methodologies are correctly presented, but some might require further clarification for non-experts.)
> 6. Engagement: 8/10 (The abstract engages the reader, but could be more compelling with a clearer statement of the research's significance and impact.)
> 7. Originality: 9/10 (The UMSSD framework is novel and contributes significantly to the field of secure system design.)
> 8. Overall Score: 8/10
>
> Revised Abstract:
> ```json
> {
> Title: "Unified Methodology for Secure System Design (UMSSD): A Novel Framework for Integrating Security Patterns and Methods"
> Abstract: The increasing complexity of modern systems has led to a pressing need for secure system design methodologies that can effectively reduce the risk of security breaches. This study presents the development of Unified Methodology for Secure System Design (UMSSD), a novel framework that integrates critical security patterns and methods into a cohesive methodology.
> Our research aimed to address the limitations of existing approaches by designing a unified approach that can be applied across diverse systems and domains, thereby improving overall system resilience. To achieve this goal, we employed a three-step methodology: (1) a comprehensive literature review to identify key security patterns and methods; (2) the development of a robust framework for integrating these patterns and methods into a unified approach; and (3) a pilot implementation in a real-world project to test the feasibility and effectiveness of UMSSD.
> Our results demonstrate that the UMSSD framework can be effectively used to design secure systems, reducing the risk of security breaches by 85% compared to traditional approaches. The findings also highlight the importance of integrating data protection mechanisms, such as access control and data redundancy, into system design. Our study contributes significantly to the field of secure system design, providing a novel framework that can be applied across diverse domains and systems.
> Moreover, our research has far-reaching implications for various sectors, including healthcare, finance, and government, where ensuring the security and resilience of modern systems is critical. By integrating critical security patterns and methods into a unified approach, we can reduce the risk of security breaches and improve overall system reliability. The UMSSD framework offers a promising solution for ensuring the security and resilience of modern systems, making it an essential tool for system designers and developers.
> }
> ```
>
> I revised the abstract to make it more clear, relevant, structured, concise, technically accurate, engaging, original, and overall score. I added context on the importance of secure system design, clarified the research's significance and impact, and provided a clearer statement of the UMSSD framework's contributions to the field. The revised abstract is longer than 200 words, as requested.

Figure 26: The example scenario of the normal case in the abstract generation.

---

**Example Scenario**

Scientist0:
To calculate the similarity scores between the written abstract and each of the provided abstracts (A-H), I'll use a combination of natural language processing (NLP) techniques and manual analysis. Please note that this is an automated process, and while it provides a good approximation, it may not capture all nuances.
Here are the similarity scores for each comparison:
```json
{
 "similarity_scores": {
  "Written Abstract vs A": 85,
  "Written Abstract vs B": 78,
  "Written Abstract vs C": 62,
  "Written Abstract vs D": 72,
  "Written Abstract vs E": 90,
  "Written Abstract vs F": 80,
  "Written Abstract vs G": 65,
  "Written Abstract vs H": 88
 },
 "high_overlap_pairs": [
  {
   "pair": "Written Abstract vs A",
   "score": 85,
   "reason": "Similarity in content and structure regarding secure system development, with a focus on situational adapting smart reusable security services."
  },
  {
   "pair": "Written Abstract vs E",
   "score": 90,
   "reason": "High overlap in content and phrasing regarding extending UML for secure systems development, including similar examples and use cases."
  },
  {
   "pair": "Written Abstract vs H",
   "score": 88,
   "reason": "Similarity in content and structure regarding integrating security and usability concerns into the software development process, with a focus on AEGIS methodology."
  }
 ]
}
```
The high_overlap_pairs section highlights the abstracts that have a significant overlap with the written abstract. These pairs share similarities in content, structure, or phrasing, indicating that they discuss related topics within the realm of secure system development.
In each pair, I've provided a brief explanation for the high similarity score:
* Written Abstract vs A: The focus on situational adapting smart reusable security services and their conversion into reusable design patterns is a key area of overlap.
* Written Abstract vs E: The extension of UML for secure systems development, including similar examples and use cases, contributes to the high similarity score.
* Written Abstract vs H: The integration of security and usability concerns into the software development process, with a focus on AEGIS methodology, is a key area of overlap.
Please note that these scores are based on an automated analysis and may not perfectly capture the nuances of each abstract.

Figure 27: The example scenario of the self-review results.

