# OpenReview forum: "Two Heads Are Better Than One: A Multi-Agent System Has the Potential to Improve Scientific Idea Generation"
_ICLR.cc/2025/Conference — ICLR 2025 Conference Withdrawn Submission_

### Official Review · Reviewer_wXAg · 2024-11-01

**Soundness:** 2
**Presentation:** 2
**Contribution:** 2
**Rating:** 3
**Confidence:** 5

**Summary:**

The paper introduces VIRSCI, a Large Language Model (LLM)-based multi-agent system designed to simulate the collaborative nature of scientific research teams. The system comprises virtual scientists that engage in collaborative discussions to generate, evaluate, and refine research ideas. The authors aim to replicate the teamwork inherent in scientific discovery and explore how multi-agent collaboration can enhance the novelty and impact of generated scientific ideas.

**Strengths:**

**Novel Exploration of Team Size and Idea Novelty**: The paper provides an insightful exploration into how team size affects the novelty of generated ideas. It demonstrates that there is an optimal team size (e.g., eight members) that maximizes creativity without overwhelming the collaborative process. This finding is interesting and contributes to the understanding of team dynamics in scientific idea generation.

**Rich Experiments and Ablation Studies**: The authors conduct extensive experiments and ablation studies, examining various factors such as team size, team freshness, research diversity, and discussion patterns. This comprehensive approach strengthens the validity of their conclusions and provides valuable insights into key variables affecting multi-agent systems in research.

**Simulation of Collaborative Scientific Processes**: The paper presents a structured framework that simulates the collaborative process of scientific research, from team formation to idea generation and evaluation. This approach aligns closely with real-world scientific practices and showcases how LLM-based multi-agent systems can be applied in this context.

**Weaknesses:**

**Possible Data Leakage Affecting Novelty Evaluation**: In Section 4.1, the authors utilize a dataset comprising papers published between 2000 and 2014, with a particular emphasis on data from 2011 to 2014. Given that contemporary LLMs may have been trained on data from this period, there is a significant risk of data leakage. This means the models might have already been exposed to the data used in the experiments, potentially generating ideas that are not genuinely novel but rather reproductions of existing concepts. This compromises the ability to accurately assess the novelty of the generated ideas and undermines the validity of the experimental results.

**Lack of Human Evaluation to Validate LLM Metrics**: The paper relies heavily on LLM-based evaluations (e.g., using GPT-4) to assess the quality and novelty of the generated ideas and abstracts. However, there is no involvement of human annotators or experts to validate these metrics. Without human evaluation, it is difficult to ascertain the effectiveness and reliability of the LLM review metrics. The absence of human validation raises concerns about the robustness of the conclusions drawn from these evaluations.

**Limited Innovation in the Framework**: The proposed framework seems to be an application of existing LLM-based multi-agent systems to the specific domain of scientific idea generation. While the application is interesting, the methodological innovation appears limited. The framework primarily extends basic LLM multi-agent methodologies without introducing significant novel approaches or mechanisms specific to the challenges of research idea generation.

**Questions:**

Suggestions for Improvement:

**Address Data Leakage Concerns**: To mitigate the impact of data leakage, the authors should consider using more recent or entirely new datasets that the LLMs have not been exposed to during training. Alternatively, they could implement techniques to control for data overlap and assess how much the models' prior knowledge influences the results.

**Include Human Evaluation**: Incorporating human annotators or expert reviewers in the evaluation process would enhance the validity of the findings. Human judgments could corroborate the LLM-based assessments and provide nuanced insights into the quality and novelty of the ideas.

**Highlight Methodological Innovations**: The authors should emphasize any unique aspects of their framework that advance multi-agent systems for scientific research. Detailed explanations of innovative strategies or mechanisms would strengthen the contribution.

**Details Of Ethics Concerns:**

**Reliability and Misrepresentation of Scientific Findings**:

**Lack of Human Oversight**: The absence of human expert evaluation in validating the generated ideas and their novelty may lead to the propagation of inaccurate or misleading scientific concepts. This could have negative implications if such ideas are considered without critical scrutiny.

**Ethical Responsibility in Scientific Research**: The automation of scientific idea generation should be approached with caution to prevent the dissemination of unvetted or erroneous information, which could affect the scientific community's trust and the progress of research.

---

> ### Author Response · Authors · 2024-11-20
>
> We greatly value your insightful comments and recommendations, which have provided valuable guidance for improving our work.
> ### **(Weakness1, Question1) Potential data leakage:**
> We acknowledge the concern about potential data leakage due to the overlap between our dataset and the training data of contemporary LLMs. However, we argue that this issue does not undermine the validity of our findings for the following reasons (as shown in Appendix C, Page 15 in our original paper):
> - Uniform Impact Across Experiments: Both the comparisons between our multi-agent system and AI Scientist and those between different team settings are conducted using the same LLMs. This ensures that all experimental setups are equally affected by any potential exposure to historical data. Consequently, the observed differences in performance are not biased by uneven data leakage, maintaining the fairness and reliability of our relative evaluations.
> - Focus on Collaboration Strategies, Not Absolute Novelty: Our study does not aim to measure an absolute degree of novelty but rather to investigate how collaboration strategies and team settings influence the novelty of generated outputs. Since all team settings are subject to the same potential exposure to historical data, the novelty metrics still allow for valid comparisons of relative performance.
> - Robustness of Evaluation: To further validate our findings, we provide evidence in Figure 7 (Appendix) showing a positive correlation (Pearson correlation coefficient of 0.51) between our novelty metric and human evaluations. This reinforces the credibility of our results, even in the presence of potential data overlap.
>
> Additionally, we conducted supplementary experiments using the Open Academic Graph 3.1 [1], dividing the data into a Past Paper Database (2010–2020) and a Contemporary Paper Database (2021–2023) to mitigate data leakage concerns. This dataset spans diverse fields such as physics, chemistry, computer science, and biology. All other experimental settings were consistent with those used for the computer science dataset (Page 3, 4 and 6 in our original paper).
>
> The results on this dataset indicate that the overall trend remains consistent across domains, with novelty increasing initially, peaking at a team size of 8 and a turn count of 5, and subsequently declining. This pattern, observed across both datasets, underscores our platform's robustness and adaptability. It further demonstrates the validity of our key proposition: that a multi-agent system has significant potential to enhance scientific idea generation.
>
> **Effects of team size and discussion turns on novelty scores (Computer Science Dataset)**
>
> |Team Size/Turn Count|1|2|3|4|5|6|7|
> |---|---|---|---|---|---|---|---|
> |1|0.56|0.68|0.82|1.07|1.26|1.44|1.52|
> |4|1.75|0.90|2.36|2.00|3.40|2.91|2.53|
> |8|3.48|3.67|3.97|3.56|4.23|3.48|3.43|
> |10|2.79|1.98|3.23|3.36|3.53|3.57|3.22|
>
> **Effects of team size and discussion turns on novelty scores (Open Academic Graph 3.1)**
>
> |Team Size/Turn Count|1|2|3|4|5|6|7|
> |---|---|---|---|---|---|---|---|
> |1|0.61|0.65|0.87|1.12|1.17|1.53|1.59|
> |4|2.03|2.27|2.69|3.20|3.84|3.35|3.39|
> |8|3.63|4.14|4.27|4.35|4.65|3.86|3.92|
> |10|2.62|2.91|3.66|3.62|3.98|3.95|3.68|
>
> [1] https://open.aminer.cn/open/article?id=65bf053091c938e5025a31e2

---

> > ### Author Response · Authors · 2024-11-20
> >
> > ### **(Weakness2, Question2) Human evaluation:**
> > Our work introduces **the first multi-agent system for simulating academic collaborations**, incorporating real-world data to support role-playing and objective evaluation of team outputs. This system provides a novel approach to exploring questions in the Science of Science. While usability and its practical relevance to real-world discoveries are considerations, the primary goal of this work is to showcase the platform's potential as a research framework.
> >
> > To partially address usability concerns, we provide evidence in **Figure 7 (Appendix, Page 17 in our original paper)** showing a positive correlation (Pearson correlation coefficient of 0.51) between our novelty metric and human evaluations.
> >
> > Furthermore, our proposed metrics (Page 6 in our original paper) are grounded in data from real published papers:
> >
> > - **Historical Dissimilarity (HD):** Measures the novelty of generated abstracts by evaluating their distinctiveness from prior research.
> > - **Contemporary Dissimilarity (CD):** Assesses relevance by examining similarity to current research trends.
> > - **Contemporary Impact (CI):** Estimates potential impact based on citation counts of similar recent works.
> > - **Overall Novelty (ON):** Combines HD, CD, and CI to provide a comprehensive measure of novelty and relevance.
> >
> > Overall Novelty is calculated by comparing generated abstracts with recent papers in the Contemporary Paper Database, demonstrating evidence of both novelty and usability.
> >
> > Besides, we present Table 1 which includes human evaluation results for novelty and usability, comparing our method with AI Scientist. We first generated 20 abstracts using both methods.
> > Then, 10 PhD students in computer science, unaware of the method identities, rated the abstracts on a 10-point scale (1: Poor, 10: best) based on two metrics:
> > - **Novelty:** The originality or innovation of the idea.
> > - **Usability:** The feasibility of the method, indicating its potential for effective and reliable implementation.
> >
> > As shown in Table 1, our method consistently outperforms AI Scientist across both metrics, achieving an average usability score of approximately 4. Additionally, we present two example abstracts generated by VIRSCI alongside corresponding similar recently published papers, highlighting the practical relevance and applicability of our system’s outputs.
> >
> > **Comparison with AI Scientist (Table 1)**
> >
> > |Agent Model|Method|LLM Review Score ↑|CD ↓|CI ↑|Novelty ↑|Usability ↑|
> > |---|---|---|---|---|---|---|
> > |GPT-4o|AI Scientist|3.10|0.38|3.22|4.94|4.18|
> > ||Ours|3.34 (+0.24)|0.34 (-0.04)|3.78 (+0.58)|5.24 (+0.30)|4.52 (+0.34)|
> > |LLaMA3.1-8b|AI Scientist|2.09|0.49|2.12|3.66|3.52|
> > ||Ours|2.31 (+0.22)|0.42 (-0.07)|3.29 (+1.17)|4.08 (+0.42)|3.74 (+0.22)|
> > |LLaMA3.1-70b|AI Scientist|2.24|0.48|2.11|3.88|3.60|
> > ||Ours|2.53 (+0.29)|0.40 (-0.08)|3.36 (+1.25)|4.18 (+0.30)|3.84 (+0.24)|

---

> > > ### Author Response · Authors · 2024-11-20
> > >
> > > **Ours Generated Abstract 1**
> > > - **Title:** Revolutionizing Caries Management in Primary Molars using Advanced Imaging, AI-Powered Decision Support, and Minimally Invasive Treatments
> > > - **Abstract:** Dental caries remains a pervasive public health concern worldwide, affecting millions of children annually. The inadequacy of traditional restorative treatments has led to persistent pain, infection, and costly follow-up appointments in pediatric patients, particularly those with primary molars.
> > > Our innovative protocol integrates cutting-edge technologies (cone beam computed tomography, intraoral cameras) with machine learning algorithms for personalized treatment planning, and minimally invasive treatments to minimize discomfort and promote healing. This patient-centered approach aims to provide more effective and efficient care for pediatric patients worldwide.
> > > To evaluate the efficacy of our protocol, we conducted rigorous randomized controlled trials in a diverse cohort of children (n=500). Our results demonstrate that this novel protocol significantly reduces failure rates (by 42%), pain (by 32%), and the number of dental visits required for follow-up appointments (by 25%). Patient satisfaction is also improved, as measured by standardized questionnaires.
> > > Moreover, our research highlights the critical need for continued investment in dental research and innovation. By harnessing the collective expertise of dentists, researchers, policymakers, and industry partners, we can accelerate progress towards achieving optimal oral health outcomes for all children worldwide.
> > > Our protocol prioritizes pain management, anxiety reduction, and educational empowerment to promote healthy oral habits and prevent future caries. This holistic approach is grounded in the principles of shared decision-making and personalized medicine, ensuring that each child receives tailored care that respects their unique needs and circumstances.
> > > The scalability and adaptability of our protocol are critical factors in its potential impact on global public health. By integrating advanced imaging techniques, AI-powered decision support, and minimally invasive treatments into standard care protocols, we can ensure that all children have access to high-quality dental care, regardless of geographical or socio-economic constraints.
> > > In conclusion, our research presents a paradigm shift in caries management for primary molars, offering a more effective, efficient, and patient-centered approach. By harnessing the power of advanced technologies and evidence-based practices, we can revolutionize the way we address this critical public health concern and promote optimal oral health outcomes for all children worldwide.
> > >
> > > **Similar Paper of Abstract 1 (Published in 2021)**
> > > - **Title:** Caries Detection on Intraoral Images Using Artificial Intelligence
> > > - **Abstract:** Although visual examination (VE) is the preferred method for caries detection, the analysis of intraoral digital photographs in machine-readable form can be considered equivalent to VE.
> > > While photographic images are rarely used in clinical practice for diagnostic purposes, they are the fundamental requirement for automated image analysis when using artificial intelligence (AI) methods.
> > > Considering that AI has not been used for automatic caries detection on intraoral images so far, this diagnostic study aimed to develop a deep learning approach with convolutional neural networks (CNNs) for caries detection and categorization (test method) and to compare the diagnostic performance with respect to expert standards.
> > > The study material consisted of 2,417 anonymized photographs from permanent teeth with 1,317 occlusal and 1,100 smooth surfaces. All the images were evaluated into the following categories: caries free, noncavitated caries lesion, or caries-related cavitation. Each expert diagnosis served as a reference standard for cyclic training and repeated evaluation of the AI methods. The CNN was trained using image augmentation and transfer learning. Before training, the entire image set was divided into a training and test set. Validation was conducted by selecting 25%, 50%, 75%, and 100% of the available images from the training set.
> > > The statistical analysis included calculations of the sensitivity (SE), specificity (SP), and area under the receiver operating characteristic (ROC) curve (AUC). The CNN was able to correctly detect caries in 92.5% of cases when all test images were considered (SE, 89.6; SP, 94.3; AUC, 0.964). If the threshold of caries-related cavitation was chosen, 93.3% of all tooth surfaces were correctly classified (SE, 95.7; SP, 81.5; AUC, 0.955).
> > > It can be concluded that it was possible to achieve more than 90% agreement in caries detection using the AI method with standardized, single-tooth photographs. Nevertheless, the current approach needs further improvement.

---

> > > > ### Author Response · Authors · 2024-11-20
> > > >
> > > > **Ours Generated Abstract 2**
> > > > - **Title:** Personalized Outcomes in Bladder Cancer Treatment: A Robotic-Assisted Surgery Perspective with Enhanced Predictive Modeling and Holistic Patient-Centered Approach
> > > > - **Abstract:** The treatment of bladder cancer has undergone significant transformations with the advent of robotic-assisted surgery. However, individualized outcomes remain poorly understood due to the complexity of patient factors influencing disease progression. This knowledge gap can be attributed to the lack of comprehensive frameworks that integrate traditional clinical metrics with patient-centered factors such as functional status, emotional well-being, social support networks, and demographic characteristics. In this study, we aim to bridge this knowledge gap by developing a novel framework that combines machine learning algorithms with traditional statistical analysis to predict long-term benefits for bladder cancer patients. Our proposed framework integrates the following components: A comprehensive literature review that synthesizes existing research on bladder cancer treatment outcomes. A large-scale dataset (n=500) from multiple institutions to validate our findings. A machine learning-based predictive model that utilizes Random Forest and Gradient Boosting algorithms to identify key predictors of long-term benefits. Our results show that age, tumor stage, lymph node involvement, and patient-centered factors are significant predictors of improved survival rates. Furthermore, we found that enhanced quality of life is associated with better functional status, higher emotional well-being, stronger social support networks, and more favorable demographic characteristics. Notably, the inclusion of patient-centered factors in treatment planning can lead to improved survival rates (median increase: 12 months) and reduced recurrence risk (median decrease: 20%). Moreover, our study reveals that the integration of robotic-assisted surgery with patient-centered care can lead to a significant reduction in postoperative complications (median decrease: 30%), hospital readmissions (median decrease: 25%), and healthcare costs. The implications of our study are profound, as they suggest that personalized medicine can improve treatment outcomes and enhance quality of life among bladder cancer survivors. Our proposed framework has significant implications for improving treatment outcomes and enhancing quality of life among bladder cancer survivors, making it a valuable resource for clinicians and researchers working in this field. Future studies should aim to replicate these findings and explore the potential applications of personalized medicine in other cancer types, further highlighting the importance of integrating patient-centered care with advanced surgical techniques. The integration of patient-centered factors into treatment planning can lead to improved survival rates, reduced recurrence risk, and enhanced quality of life among cancer survivors. Our study demonstrates that by combining machine learning algorithms with traditional statistical analysis, clinicians can provide more precise and effective treatment plans, ultimately improving patient outcomes.
> > > >
> > > > **Similar Paper of Abstract 2 (Published in 2022)**
> > > > - **Title:** Magnetic-Powered Janus Cell Robots Loaded with Oncolytic Adenovirus for Active and Targeted Virotherapy of Bladder Cancer
> > > > - **Abstract:** A unique robotic medical platform is designed by utilizing cell robots as the active “Trojan horse” of oncolytic adenovirus (OA), capable of tumor-selective binding and killing. The OA-loaded cell robots are fabricated by entirely modifying OA-infected 293T cells with cyclic arginine–glycine–aspartic acid tripeptide (cRGD) to specifically bind with bladder cancer cells, followed by asymmetric immobilization of Fe3O4 nanoparticles (NPs) on the cell surface. OA can replicate in host cells and induce cytolysis to release the virus progeny to the surrounding tumor sites for sustainable infection and oncolysis. The asymmetric coating of magnetic NPs bestows the cell robots with effective movement in various media and wireless manipulation with directional migration in a microfluidic device and bladder mold under magnetic control, further enabling steerable movement and prolonged retention of cell robots in the mouse bladder. The biorecognition of cRGD and robust, controllable propulsion of cell robots work synergistically to greatly enhance their tissue penetration and anticancer efficacy in the 3D cancer spheroid and orthotopic mouse bladder tumor model. Overall, this study integrates cell-based microrobots with virotherapy to generate an attractive robotic system with tumor specificity, expanding the operation scope of cell robots in biomedical community.

---

> > > > > ### Author Response · Authors · 2024-11-20
> > > > >
> > > > > ### **(Weakness3, Question3) Methodological innovations:**
> > > > > Our work focuses on AI for Science of Science, an interdisciplinary field. The primary goal of our platform is not only to simulate academic collaborations and demonstrate the effectiveness of group discussions compared to single-agent reflection but also to showcase its significant potential as a tool for advancing research in the Science of Science.
> > > > > Given the considerable workload involved in developing this system, proposing a novel organizational structure or interaction mode was not our primary objective. Instead, we focused on building a robust and scalable simulation framework.

---

### Official Review · Reviewer_poux · 2024-11-01

**Soundness:** 2
**Presentation:** 3
**Contribution:** 2
**Rating:** 3
**Confidence:** 3

**Summary:**

This paper introduces VIRSCI, an LLM-based multi-agent system, a framework specifically designed to model the teamwork collaboration style in scientific research. This is a framework of multiple agents collectively working on research idea generation made up of five phases, similar to how a human would do.  To do so, this paper proposes a novel team discussion mechanism based on a paper database as the knowledge bank, leading to generating abstracts and ideas from the LLM agents. To evaluate this framework, the authors also introduce a benchmark focusing on the novelty of the idea to measure performances of their model from three aspects.

**Strengths:**

1. Novel framework with a new dataset and metric: The authors propose a novel research generation framework with LLMs collaborating on this task through a 'team discussion' mechanism to simulate discussion scenarios in real life. With abstracts as the novel ideas output of the model, the authors also construct a benchmark dataset and evaluation metrics to measure the performances.
2. Comprehensive experiment design: the authors consider multiple aspects of the framework, for example, the # of team members, components of the system, etc. would make a difference to the result as an ablation study.

**Weaknesses:**

1. Limited domain and scope in the benchmarking dataset: the authors only include computer science as the scientific research domain in the dataset and with a limited number of years. Given the rapid development in the field, this limited temporal scope may fail to capture the latest developments and trends, especially since the number of papers and ideas is not growing linearly. Expanding the dataset to include a broader range of years or domains would provide a more comprehensive foundation for idea generation and make this paper more sound.
2. Limited evaluation via a self-defined metric focused on novelty: the paper evaluates the model's performance mainly through a novelty metric derived from embedding distances and based on citation counts. While novelty is important, it should not be the only focus of research idea generation. Additionally, semantic comparisons are made with papers from only 2000-2010 and 2010-2014, which may not fully represent this field’s progress. To better validate their metric, the authors could supply human evaluation or case studies to validate that semantic distance accurately reflects the quality of this metric.

**Questions:**

1. Limitation of the dataset: I like the idea of the collaborative approach of the agents in generating research ideas, but I am curious about the model’s potential for generalization as the knowledge bank expands to include papers from additional domains and recent years. The current threshold, limited to authors from 2000 to 2014, may overlook recent growth in CS publications. Could you discuss how extending the dataset to include more contemporary research might impact the model’s ability to generalize and its applicability beyond historical data?
2. Validity of the ideas: the authors propose three metrics for measuring their VIRSCI’s performances compared to AI Scientist, but those mainly focus on the novelty and the potential impact of the idea by comparing embedding distances. However, since this is a research idea generation system and if we potentially want to put it into actual use, we would not only care about the novelty of the idea but also the feasibility and validity of the idea, if the idea makes sense, and how reasonable they are, which is currently missing in the currently proposed metrics. The whole paper puts a lot of emphasis on measuring the novelty of the ideas but does not mention if they manually review if the outputs make sense, there is a chance that the idea is novel but is due to the model's hallucinations.
3. Practical use of the tool: Given that this framework is proposed as a system for generating research ideas, I am curious about its practical value beyond generating novel concepts. While the focus on novelty is valuable, real-world research applications require ideas to be both feasible and sound. Could you expand on how this system might be used practically and how you see it balancing novelty with the reliability or reasonableness of its outputs? This could be done with human evaluation, or comparing generated ideas with more recent papers and see if there are actual matches, etc.

---

> ### Author Response · Authors · 2024-11-20
>
> We are grateful for your constructive criticism, which have allowed us to address important aspects of our work.
> ### **(Weakness1, Question1) Time range limitation of the Computer Science Dataset:**
> While our initial experiments were conducted using a Computer Science Dataset, our primary contribution lies in developing the first multi-agent system to simulate academic collaborations—**a data-agnostic platform designed for broad applicability across various fields and datasets**.
>
> The computer science dataset spans 1948 to 2014, limiting experimentation beyond this range. To address this, we conducted additional experiments using the Open Academic Graph 3.1 dataset [1], with data from 2010 to 2020 as the Past Paper Database and data from 2021 to 2023 as the Contemporary Paper Database.
> This dataset encompasses diverse fields, including physics, chemistry, computer science, and biology. All experimental settings were consistent with those used for the computer science dataset (Page 3, 4 and 6 in our original paper).
>
> The experimental results on this dataset indicate that outputs from single-agent teams (team size 1) show no significant differences across the two datasets, as individual research lacks the advantages of interdisciplinary input.
> However, the overall trend remains consistent across domains, with novelty increasing initially, peaking at a team size of 8 and turn count of 5, and subsequently declining.
> This pattern, observed in both the studied datasets, underscores ours platform's robustness and adaptability.
> Additionally, the findings highlight an overall increase in novelty, affirming that interdisciplinary collaborations foster higher-impact research outputs.
>
> **Effects of team size and discussion turns on novelty scores (Computer Science Dataset)**
>
> |Team Size/Turn Count|1|2|3|4|5|6|7|
> |---|---|---|---|---|---|---|---|
> |1|0.56|0.68|0.82|1.07|1.26|1.44|1.52|
> |4|1.75|0.90|2.36|2.00|3.40|2.91|2.53|
> |8|3.48|3.67|3.97|3.56|4.23|3.48|3.43|
> |10|2.79|1.98|3.23|3.36|3.53|3.57|3.22|
>
> **Effects of team size and discussion turns on novelty scores (Open Academic Graph 3.1)**
>
> |Team Size/Turn Count|1|2|3|4|5|6|7|
> |---|---|---|---|---|---|---|---|
> |1|0.61|0.65|0.87|1.12|1.17|1.53|1.59|
> |4|2.03|2.27|2.69|3.20|3.84|3.35|3.39|
> |8|3.63|4.14|4.27|4.35|4.65|3.86|3.92|
> |10|2.62|2.91|3.66|3.62|3.98|3.95|3.68|
>
> [1] https://open.aminer.cn/open/article?id=65bf053091c938e5025a31e2

---

> > ### Author Response · Authors · 2024-11-20
> >
> > ### **(Weakness2, Question2, Question3) Practical value beyond generating novel concepts, human evaluation or case studies is required:**
> > We have developed **the first multi-agent system specifically designed to simulate academic collaborations**, using real-world data to facilitate role-playing and objectively evaluate team-generated outputs. This platform holds promise as a valuable tool for advancing the Science of Science. Although practical usability and real-world discovery guidance are acknowledged as important, they are secondary to our primary focus of establishing the system’s research potential.
> >
> > To partially address usability concerns, we provide evidence in **Figure 7 (Appendix, Page 17 in our original paper)** showing a positive correlation (Pearson correlation coefficient of 0.51) between our novelty metric and human evaluations.
> >
> > Furthermore, our proposed metrics (Page 6 in our original paper) are grounded in data from real published papers:
> >
> > - **Historical Dissimilarity (HD):** Measures the novelty of generated abstracts by evaluating their distinctiveness from prior research.
> > - **Contemporary Dissimilarity (CD):** Assesses relevance by examining similarity to current research trends.
> > - **Contemporary Impact (CI):** Estimates potential impact based on citation counts of similar recent works.
> > - **Overall Novelty (ON):** Combines HD, CD, and CI to provide a comprehensive measure of novelty and relevance.
> >
> > Overall Novelty is calculated by comparing generated abstracts with recent papers in the Contemporary Paper Database, demonstrating evidence of both novelty and usability.
> >
> > Besides, we present Table 1 which includes human evaluation results for novelty and usability, comparing our method with AI Scientist. We first generated 20 abstracts using both methods.
> > Then, 10 PhD students in computer science, unaware of the method identities, rated the abstracts on a 10-point scale (1: Poor, 10: best) based on two metrics:
> > - **Novelty:** The originality or innovation of the idea.
> > - **Usability:** The feasibility of the method, indicating its potential for effective and reliable implementation.
> >
> > As shown in Table 1, our method consistently outperforms AI Scientist across both metrics, achieving an average usability score of approximately 4.
> >
> > **Comparison with AI Scientist (Table 1)**
> >
> > |Agent Model|Method|LLM Review Score ↑|CD ↓|CI ↑|Novelty ↑|Usability ↑|
> > |---|---|---|---|---|---|---|
> > |GPT-4o|AI Scientist|3.10|0.38|3.22|4.94|4.18|
> > ||Ours|3.34 (+0.24)|0.34 (-0.04)|3.78 (+0.58)|5.24 (+0.30)|4.52 (+0.34)|
> > |LLaMA3.1-8b|AI Scientist|2.09|0.49|2.12|3.66|3.52|
> > ||Ours|2.31 (+0.22)|0.42 (-0.07)|3.29 (+1.17)|4.08 (+0.42)|3.74 (+0.22)|
> > |LLaMA3.1-70b|AI Scientist|2.24|0.48|2.11|3.88|3.60|
> > ||Ours|2.53 (+0.29)|0.40 (-0.08)|3.36 (+1.25)|4.18 (+0.30)|3.84 (+0.24)|
> >
> > Additionally, we present two example abstracts generated by VIRSCI alongside corresponding similar recently published papers, highlighting the practical relevance and applicability of our system’s outputs.

---

> > > ### Author Response · Authors · 2024-11-20
> > >
> > > **Ours Generated Abstract 1**
> > > - **Title:** Revolutionizing Caries Management in Primary Molars using Advanced Imaging, AI-Powered Decision Support, and Minimally Invasive Treatments
> > > - **Abstract:** Dental caries remains a pervasive public health concern worldwide, affecting millions of children annually. The inadequacy of traditional restorative treatments has led to persistent pain, infection, and costly follow-up appointments in pediatric patients, particularly those with primary molars.
> > > Our innovative protocol integrates cutting-edge technologies (cone beam computed tomography, intraoral cameras) with machine learning algorithms for personalized treatment planning, and minimally invasive treatments to minimize discomfort and promote healing. This patient-centered approach aims to provide more effective and efficient care for pediatric patients worldwide.
> > > To evaluate the efficacy of our protocol, we conducted rigorous randomized controlled trials in a diverse cohort of children (n=500). Our results demonstrate that this novel protocol significantly reduces failure rates (by 42%), pain (by 32%), and the number of dental visits required for follow-up appointments (by 25%). Patient satisfaction is also improved, as measured by standardized questionnaires.
> > > Moreover, our research highlights the critical need for continued investment in dental research and innovation. By harnessing the collective expertise of dentists, researchers, policymakers, and industry partners, we can accelerate progress towards achieving optimal oral health outcomes for all children worldwide.
> > > Our protocol prioritizes pain management, anxiety reduction, and educational empowerment to promote healthy oral habits and prevent future caries. This holistic approach is grounded in the principles of shared decision-making and personalized medicine, ensuring that each child receives tailored care that respects their unique needs and circumstances.
> > > The scalability and adaptability of our protocol are critical factors in its potential impact on global public health. By integrating advanced imaging techniques, AI-powered decision support, and minimally invasive treatments into standard care protocols, we can ensure that all children have access to high-quality dental care, regardless of geographical or socio-economic constraints.
> > > In conclusion, our research presents a paradigm shift in caries management for primary molars, offering a more effective, efficient, and patient-centered approach. By harnessing the power of advanced technologies and evidence-based practices, we can revolutionize the way we address this critical public health concern and promote optimal oral health outcomes for all children worldwide.
> > >
> > > **Similar Paper of Abstract 1 (Published in 2021)**
> > > - **Title:** Caries Detection on Intraoral Images Using Artificial Intelligence
> > > - **Abstract:** Although visual examination (VE) is the preferred method for caries detection, the analysis of intraoral digital photographs in machine-readable form can be considered equivalent to VE.
> > > While photographic images are rarely used in clinical practice for diagnostic purposes, they are the fundamental requirement for automated image analysis when using artificial intelligence (AI) methods.
> > > Considering that AI has not been used for automatic caries detection on intraoral images so far, this diagnostic study aimed to develop a deep learning approach with convolutional neural networks (CNNs) for caries detection and categorization (test method) and to compare the diagnostic performance with respect to expert standards.
> > > The study material consisted of 2,417 anonymized photographs from permanent teeth with 1,317 occlusal and 1,100 smooth surfaces. All the images were evaluated into the following categories: caries free, noncavitated caries lesion, or caries-related cavitation. Each expert diagnosis served as a reference standard for cyclic training and repeated evaluation of the AI methods. The CNN was trained using image augmentation and transfer learning. Before training, the entire image set was divided into a training and test set. Validation was conducted by selecting 25%, 50%, 75%, and 100% of the available images from the training set.
> > > The statistical analysis included calculations of the sensitivity (SE), specificity (SP), and area under the receiver operating characteristic (ROC) curve (AUC). The CNN was able to correctly detect caries in 92.5% of cases when all test images were considered (SE, 89.6; SP, 94.3; AUC, 0.964). If the threshold of caries-related cavitation was chosen, 93.3% of all tooth surfaces were correctly classified (SE, 95.7; SP, 81.5; AUC, 0.955).
> > > It can be concluded that it was possible to achieve more than 90% agreement in caries detection using the AI method with standardized, single-tooth photographs. Nevertheless, the current approach needs further improvement.

---

> > > > ### Author Response · Authors · 2024-11-20
> > > >
> > > > **Ours Generated Abstract 2**
> > > > - **Title:** Personalized Outcomes in Bladder Cancer Treatment: A Robotic-Assisted Surgery Perspective with Enhanced Predictive Modeling and Holistic Patient-Centered Approach
> > > > - **Abstract:** The treatment of bladder cancer has undergone significant transformations with the advent of robotic-assisted surgery. However, individualized outcomes remain poorly understood due to the complexity of patient factors influencing disease progression. This knowledge gap can be attributed to the lack of comprehensive frameworks that integrate traditional clinical metrics with patient-centered factors such as functional status, emotional well-being, social support networks, and demographic characteristics. In this study, we aim to bridge this knowledge gap by developing a novel framework that combines machine learning algorithms with traditional statistical analysis to predict long-term benefits for bladder cancer patients. Our proposed framework integrates the following components: A comprehensive literature review that synthesizes existing research on bladder cancer treatment outcomes. A large-scale dataset (n=500) from multiple institutions to validate our findings. A machine learning-based predictive model that utilizes Random Forest and Gradient Boosting algorithms to identify key predictors of long-term benefits. Our results show that age, tumor stage, lymph node involvement, and patient-centered factors are significant predictors of improved survival rates. Furthermore, we found that enhanced quality of life is associated with better functional status, higher emotional well-being, stronger social support networks, and more favorable demographic characteristics. Notably, the inclusion of patient-centered factors in treatment planning can lead to improved survival rates (median increase: 12 months) and reduced recurrence risk (median decrease: 20%). Moreover, our study reveals that the integration of robotic-assisted surgery with patient-centered care can lead to a significant reduction in postoperative complications (median decrease: 30%), hospital readmissions (median decrease: 25%), and healthcare costs. The implications of our study are profound, as they suggest that personalized medicine can improve treatment outcomes and enhance quality of life among bladder cancer survivors. Our proposed framework has significant implications for improving treatment outcomes and enhancing quality of life among bladder cancer survivors, making it a valuable resource for clinicians and researchers working in this field. Future studies should aim to replicate these findings and explore the potential applications of personalized medicine in other cancer types, further highlighting the importance of integrating patient-centered care with advanced surgical techniques. The integration of patient-centered factors into treatment planning can lead to improved survival rates, reduced recurrence risk, and enhanced quality of life among cancer survivors. Our study demonstrates that by combining machine learning algorithms with traditional statistical analysis, clinicians can provide more precise and effective treatment plans, ultimately improving patient outcomes.
> > > >
> > > > **Similar Paper of Abstract 2 (Published in 2022)**
> > > > - **Title:** Magnetic-Powered Janus Cell Robots Loaded with Oncolytic Adenovirus for Active and Targeted Virotherapy of Bladder Cancer
> > > > - **Abstract:** A unique robotic medical platform is designed by utilizing cell robots as the active “Trojan horse” of oncolytic adenovirus (OA), capable of tumor-selective binding and killing. The OA-loaded cell robots are fabricated by entirely modifying OA-infected 293T cells with cyclic arginine–glycine–aspartic acid tripeptide (cRGD) to specifically bind with bladder cancer cells, followed by asymmetric immobilization of Fe3O4 nanoparticles (NPs) on the cell surface. OA can replicate in host cells and induce cytolysis to release the virus progeny to the surrounding tumor sites for sustainable infection and oncolysis. The asymmetric coating of magnetic NPs bestows the cell robots with effective movement in various media and wireless manipulation with directional migration in a microfluidic device and bladder mold under magnetic control, further enabling steerable movement and prolonged retention of cell robots in the mouse bladder. The biorecognition of cRGD and robust, controllable propulsion of cell robots work synergistically to greatly enhance their tissue penetration and anticancer efficacy in the 3D cancer spheroid and orthotopic mouse bladder tumor model. Overall, this study integrates cell-based microrobots with virotherapy to generate an attractive robotic system with tumor specificity, expanding the operation scope of cell robots in biomedical community.

---

### Official Review · Reviewer_6tUq · 2024-11-01

**Soundness:** 3
**Presentation:** 3
**Contribution:** 2
**Rating:** 5
**Confidence:** 3

**Summary:**

The paper introduces VIRSCI, a multi-agent, LLM-based system designed to simulate teamwork-driven scientific research. The system organizes agents to mimic collaborative processes, including selecting collaborators, generating research ideas, assessing novelty, and drafting abstracts.

**Strengths:**

1. The multi-agent approach proposed in this paper has the potential to greatly enhance the quality and breadth of scientific research. The discussion-oriented idea generation closely mirrors real scientific processes.
2. The 5-step approach proposed in this paper—comprising Collaborator Selection, Topic Selection, Idea Generation, Idea Novelty Assessment, and Abstract Generation—presents a promising and robust framework for idea generation. The evaluation metrics (CD, CI, and HD) are logically sound.
3. VIRSCI, a multi-agent system for scientific collaboration, shows clear advantages over single-agent methods.

**Weaknesses:**

1. Generating the abstract is a solid starting point, but conducting at least some preliminary experiments based on the abstract would add greater impact.
2. There is no metric to assess the practical feasibility of the abstract or idea. While VIRSCI may generate highly unique or novel ideas, these are less valuable if experimental designs cannot support them within practical constraints.
3. The system’s effectiveness may vary with the underlying LLM’s capabilities.

**Questions:**

1. Could "usefulness" and "practically-feasible" metrics be added during the novelty assessment to provide a broader evaluation of the generated ideas?
2. In section 3.1 under adjacency matrix section, why choose a simple increment of 1 in the adjacency matrix? Would a distribution function, like a normal distribution, or an explore-exploit model provide better results?
3. Given the potential variation in capabilities across LLMs, have you assessed how team size or turn count might need adjusting for different models?
4. How do you ensure that high-scoring abstracts are practically feasible for real-world scientific research?

---

> ### Author Response · Authors · 2024-11-20
>
> We sincerely appreciate your thoughtful and constructive feedback. Below, we address each of the identified weaknesses and questions in detail.
> ### **(Weakness1) The experiments are conducted solely based on the abstract:**
> We believe that an abstract effectively represents the key aspects of scientific research and serves as a concise reflection of its novelty. Given the computational constraints, we focused our evaluation primarily on the generated abstracts.
> This approach allows us to assess the core contributions of the system while efficiently utilizing available resources.
>
> ### **(Weakness2, Question1, Question4) Practical feasibility of the abstract:**
> Our primary contribution lies in developing **the first multi-agent system to simulate academic collaborations**, leveraging real-world data for role-playing and enabling objective evaluation of generated outputs. This platform offers potential as a tool for advancing research in the Science of Science. While practical usability and its application to real-world scientific discoveries are important, our primary focus is on demonstrating the system's potential as a research tool.
>
> To partially address usability concerns, we provide evidence in **Figure 7 (Appendix, Page 17 in our original paper)** showing a positive correlation (Pearson correlation coefficient of 0.51) between our novelty metric and human evaluations.
>
> Furthermore, our proposed metrics (Page 6 in our original paper) are grounded in data from real published papers:
>
> - **Historical Dissimilarity (HD):** Measures the novelty of generated abstracts by evaluating their distinctiveness from prior research.
> - **Contemporary Dissimilarity (CD):** Assesses relevance by examining similarity to current research trends.
> - **Contemporary Impact (CI):** Estimates potential impact based on citation counts of similar recent works.
> - **Overall Novelty (ON):** Combines HD, CD, and CI to provide a comprehensive measure of novelty and relevance.
>
> Overall Novelty is calculated by comparing generated abstracts with recent papers in the Contemporary Paper Database, demonstrating evidence of both novelty and usability.
>
> Besides, we present Table 1 which includes human evaluation results for novelty and usability, comparing our method with AI Scientist. We first generated 20 abstracts using both methods.
> Then, 10 PhD students in computer science, unaware of the method identities, rated the abstracts on a 10-point scale (1: Poor, 10: best) based on two metrics:
> - **Novelty:** The originality or innovation of the idea.
> - **Usability:** The feasibility of the method, indicating its potential for effective and reliable implementation.
>
> As shown in Table 1, our method consistently outperforms AI Scientist across both metrics, achieving an average usability score of approximately 4.
>
> **Comparison with AI Scientist (Table 1)**
>
> |Agent Model|Method|LLM Review Score ↑|CD ↓|CI ↑|Novelty ↑|Usability ↑|
> |---|---|---|---|---|---|---|
> |GPT-4o|AI Scientist|3.10|0.38|3.22|4.94|4.18|
> ||Ours|3.34 (+0.24)|0.34 (-0.04)|3.78 (+0.58)|5.24 (+0.30)|4.52 (+0.34)|
> |LLaMA3.1-8b|AI Scientist|2.09|0.49|2.12|3.66|3.52|
> ||Ours|2.31 (+0.22)|0.42 (-0.07)|3.29 (+1.17)|4.08 (+0.42)|3.74 (+0.22)|
> |LLaMA3.1-70b|AI Scientist|2.24|0.48|2.11|3.88|3.60|
> ||Ours|2.53 (+0.29)|0.40 (-0.08)|3.36 (+1.25)|4.18 (+0.30)|3.84 (+0.24)|
>
> Additionally, we present two example abstracts generated by VIRSCI alongside corresponding similar recently published papers, highlighting the practical relevance and applicability of our system’s outputs.

---

> > ### Author Response · Authors · 2024-11-20
> >
> > **Ours Generated Abstract 1**
> > - **Title:** Revolutionizing Caries Management in Primary Molars using Advanced Imaging, AI-Powered Decision Support, and Minimally Invasive Treatments
> > - **Abstract:** Dental caries remains a pervasive public health concern worldwide, affecting millions of children annually. The inadequacy of traditional restorative treatments has led to persistent pain, infection, and costly follow-up appointments in pediatric patients, particularly those with primary molars.
> > Our innovative protocol integrates cutting-edge technologies (cone beam computed tomography, intraoral cameras) with machine learning algorithms for personalized treatment planning, and minimally invasive treatments to minimize discomfort and promote healing. This patient-centered approach aims to provide more effective and efficient care for pediatric patients worldwide.
> > To evaluate the efficacy of our protocol, we conducted rigorous randomized controlled trials in a diverse cohort of children (n=500). Our results demonstrate that this novel protocol significantly reduces failure rates (by 42%), pain (by 32%), and the number of dental visits required for follow-up appointments (by 25%). Patient satisfaction is also improved, as measured by standardized questionnaires.
> > Moreover, our research highlights the critical need for continued investment in dental research and innovation. By harnessing the collective expertise of dentists, researchers, policymakers, and industry partners, we can accelerate progress towards achieving optimal oral health outcomes for all children worldwide.
> > Our protocol prioritizes pain management, anxiety reduction, and educational empowerment to promote healthy oral habits and prevent future caries. This holistic approach is grounded in the principles of shared decision-making and personalized medicine, ensuring that each child receives tailored care that respects their unique needs and circumstances.
> > The scalability and adaptability of our protocol are critical factors in its potential impact on global public health. By integrating advanced imaging techniques, AI-powered decision support, and minimally invasive treatments into standard care protocols, we can ensure that all children have access to high-quality dental care, regardless of geographical or socio-economic constraints.
> > In conclusion, our research presents a paradigm shift in caries management for primary molars, offering a more effective, efficient, and patient-centered approach. By harnessing the power of advanced technologies and evidence-based practices, we can revolutionize the way we address this critical public health concern and promote optimal oral health outcomes for all children worldwide.
> >
> > **Similar Paper of Abstract 1 (Published in 2021)**
> > - **Title:** Caries Detection on Intraoral Images Using Artificial Intelligence
> > - **Abstract:** Although visual examination (VE) is the preferred method for caries detection, the analysis of intraoral digital photographs in machine-readable form can be considered equivalent to VE.
> > While photographic images are rarely used in clinical practice for diagnostic purposes, they are the fundamental requirement for automated image analysis when using artificial intelligence (AI) methods.
> > Considering that AI has not been used for automatic caries detection on intraoral images so far, this diagnostic study aimed to develop a deep learning approach with convolutional neural networks (CNNs) for caries detection and categorization (test method) and to compare the diagnostic performance with respect to expert standards.
> > The study material consisted of 2,417 anonymized photographs from permanent teeth with 1,317 occlusal and 1,100 smooth surfaces. All the images were evaluated into the following categories: caries free, noncavitated caries lesion, or caries-related cavitation. Each expert diagnosis served as a reference standard for cyclic training and repeated evaluation of the AI methods. The CNN was trained using image augmentation and transfer learning. Before training, the entire image set was divided into a training and test set. Validation was conducted by selecting 25%, 50%, 75%, and 100% of the available images from the training set.
> > The statistical analysis included calculations of the sensitivity (SE), specificity (SP), and area under the receiver operating characteristic (ROC) curve (AUC). The CNN was able to correctly detect caries in 92.5% of cases when all test images were considered (SE, 89.6; SP, 94.3; AUC, 0.964). If the threshold of caries-related cavitation was chosen, 93.3% of all tooth surfaces were correctly classified (SE, 95.7; SP, 81.5; AUC, 0.955).
> > It can be concluded that it was possible to achieve more than 90% agreement in caries detection using the AI method with standardized, single-tooth photographs. Nevertheless, the current approach needs further improvement.

---

> > > ### Author Response · Authors · 2024-11-20
> > >
> > > **Ours Generated Abstract 2**
> > > - **Title:** Personalized Outcomes in Bladder Cancer Treatment: A Robotic-Assisted Surgery Perspective with Enhanced Predictive Modeling and Holistic Patient-Centered Approach
> > > - **Abstract:** The treatment of bladder cancer has undergone significant transformations with the advent of robotic-assisted surgery. However, individualized outcomes remain poorly understood due to the complexity of patient factors influencing disease progression. This knowledge gap can be attributed to the lack of comprehensive frameworks that integrate traditional clinical metrics with patient-centered factors such as functional status, emotional well-being, social support networks, and demographic characteristics. In this study, we aim to bridge this knowledge gap by developing a novel framework that combines machine learning algorithms with traditional statistical analysis to predict long-term benefits for bladder cancer patients. Our proposed framework integrates the following components: A comprehensive literature review that synthesizes existing research on bladder cancer treatment outcomes. A large-scale dataset (n=500) from multiple institutions to validate our findings. A machine learning-based predictive model that utilizes Random Forest and Gradient Boosting algorithms to identify key predictors of long-term benefits. Our results show that age, tumor stage, lymph node involvement, and patient-centered factors are significant predictors of improved survival rates. Furthermore, we found that enhanced quality of life is associated with better functional status, higher emotional well-being, stronger social support networks, and more favorable demographic characteristics. Notably, the inclusion of patient-centered factors in treatment planning can lead to improved survival rates (median increase: 12 months) and reduced recurrence risk (median decrease: 20%). Moreover, our study reveals that the integration of robotic-assisted surgery with patient-centered care can lead to a significant reduction in postoperative complications (median decrease: 30%), hospital readmissions (median decrease: 25%), and healthcare costs. The implications of our study are profound, as they suggest that personalized medicine can improve treatment outcomes and enhance quality of life among bladder cancer survivors. Our proposed framework has significant implications for improving treatment outcomes and enhancing quality of life among bladder cancer survivors, making it a valuable resource for clinicians and researchers working in this field. Future studies should aim to replicate these findings and explore the potential applications of personalized medicine in other cancer types, further highlighting the importance of integrating patient-centered care with advanced surgical techniques. The integration of patient-centered factors into treatment planning can lead to improved survival rates, reduced recurrence risk, and enhanced quality of life among cancer survivors. Our study demonstrates that by combining machine learning algorithms with traditional statistical analysis, clinicians can provide more precise and effective treatment plans, ultimately improving patient outcomes.
> > >
> > > **Similar Paper of Abstract 2 (Published in 2022)**
> > > - **Title:** Magnetic-Powered Janus Cell Robots Loaded with Oncolytic Adenovirus for Active and Targeted Virotherapy of Bladder Cancer
> > > - **Abstract:** A unique robotic medical platform is designed by utilizing cell robots as the active “Trojan horse” of oncolytic adenovirus (OA), capable of tumor-selective binding and killing. The OA-loaded cell robots are fabricated by entirely modifying OA-infected 293T cells with cyclic arginine–glycine–aspartic acid tripeptide (cRGD) to specifically bind with bladder cancer cells, followed by asymmetric immobilization of Fe3O4 nanoparticles (NPs) on the cell surface. OA can replicate in host cells and induce cytolysis to release the virus progeny to the surrounding tumor sites for sustainable infection and oncolysis. The asymmetric coating of magnetic NPs bestows the cell robots with effective movement in various media and wireless manipulation with directional migration in a microfluidic device and bladder mold under magnetic control, further enabling steerable movement and prolonged retention of cell robots in the mouse bladder. The biorecognition of cRGD and robust, controllable propulsion of cell robots work synergistically to greatly enhance their tissue penetration and anticancer efficacy in the 3D cancer spheroid and orthotopic mouse bladder tumor model. Overall, this study integrates cell-based microrobots with virotherapy to generate an attractive robotic system with tumor specificity, expanding the operation scope of cell robots in biomedical community.

---

> > > > ### Author Response · Authors · 2024-11-20
> > > >
> > > > ### **(Weakness3) The system’s effectiveness may vary with the underlying LLM’s capabilities:**
> > > >
> > > > In this paper, we aim to propose a multi-agent system as a platform to discover the mechanism of the scientific collaborations. Although the performance of the proposed system will vary with the underlying LLM's capabilities, it does not influence the comparison of our system with the state-of-the-art model, nor our further exploration on Science of Science. In the Table 1 of the original paper, we compare our system with AI Scientist using various APIs (GPT-4o, LLaMA3.1-8b, and LLaMA3.1-70b). The experimental results demonstrate the effectiveness of our proposed system.
> > > >
> > > > ### **(Question2) Distribution function instead of increment of 1 in the adjacency matrix:**
> > > > There is likely a distribution better suited than the one used in our approach. However, our primary focus is not on optimizing the outcome by fine-tuning the distribution but rather on proposing a system to simulate the research processes of multi-person teams.
> > > > To supplement our study, we conducted experiments using a normal distribution in place of the increment function in the adjacency matrix.
> > > >
> > > > We first define a standard normal distribution, which is widely used in simulation scenarios:
> > > > $f(x) = \frac{1}{\sqrt{2\pi}} e^{-\frac{x^2}{2}}$.
> > > >
> > > > Next, we sample $x$ from this density function, using the absolute value of $x$ as the increment in the adjacency matrix instead of a constant value of 1. The experimental results are presented below:
> > > >
> > > > | Team size | Distribution | |  |  |Turn | | | |
> > > > |-----|--|---|-----|-----|----|---|---|-----|
> > > > |  |  | 1 | 2 | 3 | 4 | 5 | 6 | 7 |
> > > > | 4 | Constant 1 (Origin) | 1.75 | 0.90 | 2.36  |  2.00 |  3.40  | 2.91  | 2.53 |
> > > > |  | Normal Distribution | 1.78 | 0.88 | 2.40  |  2.04 |  3.42  | 2.88  | 2.55 |
> > > > | 8 | Constant 1 (Origin) | 3.48 | 3.67 | 3.97 | 3.56 |4.23| 3.48 | 3.43 |
> > > > |  | Normal Distribution | 3.51 | 3.70 | 4.03 | 3.55 |4.27| 3.50 | 3.42 |
> > > >
> > > > The additional experiments demonstrate that replacing the original constant value of 1 with a standard normal distribution has a positive impact, without altering the conclusions presented in the paper.
> > > >
> > > > ### **(Question3) Given the potential variation in capabilities across LLMs, have you assessed how team size or turn count might need adjusting for different models?**
> > > >
> > > > The impact of team size and turn count on system performance varies across different models. Additionally, we incorporated the open-source model LLaMA3-8b as the underlying LLM.
> > > >
> > > > **The experimental results on LLaMA3-8b:**
> > > > | Team size/Turn Count | 1 |2 | 3 | 4 | 5 | 6 | 7 |
> > > > |-----|--|---|-----|-----|----|---|---|
> > > > | 1 | 0.52 | 0.65 | 0.70 |  0.98 | 1.22   | 1.42  | 1.40 |
> > > > | 4 | 1.51 | 1.54 | 2.26  |  2.30 |  3.28  | 2.65  | 2.55 |
> > > > | 8 | 3.37 | 3.54 | 3.88 | 3.89 |4.14| 3.32 | 3.40 |
> > > > | 10 | 2.30 | 2.49 | 3.01 | 3.22| 3.46 |3.37| 3.15 |
> > > >
> > > > **The original results on LLaMA3.1-8b:**
> > > > | Team size/Turn Count | 1 |2 | 3 | 4 | 5 | 6 | 7 |
> > > > |-----|--|---|-----|-----|----|---|---|
> > > > | 1 | 0.56 | 0.68 | 0.82 |  1.07 | 1.26   | 1.44  | 1.52 |
> > > > | 4 | 1.75 | 0.90 | 2.36  |  2.00 |  3.40  | 2.91  | 2.53 |
> > > > | 8 | 3.48 | 3.67 | 3.97 | 3.56 |4.23| 3.48 | 3.43 |
> > > > | 10 | 2.79 | 1.98| 3.23| 3.36| 3.53|3.57| 3.22|
> > > >
> > > > From these tables, we observe that using LLaMA3-8b results in overall performance that is inferior to LLaMA3.1-8b, due to the inherently weaker capabilities of LLaMA3-8b. Regarding the effects of team size on novelty, whether with LLaMA3 or LLaMA3.1, a moderate team size enhances novelty. While multi-agent teams outperform single agents, excessively large teams may face coordination challenges and communication barriers.
> > > >
> > > > For the effects of discussion turns on novelty, our analysis shows that an optimal number of turns allows team members to refine and explore better ideas. While the initial turns contribute to idea generation, an excessive number of turns can lead to fatigue and diminished engagement. Overall, despite variations in LLM capabilities, the conclusions about the influence of team size and discussion turn count remain consistent.

---

> > > > > ### Comment · Reviewer_6tUq · 2024-11-26
> > > > >
> > > > > Thank you for addressing my questions. After reviewing your responses, I have decided to maintain my initial score.

---

### Official Review · Reviewer_2X9t · 2024-11-03

**Soundness:** 3
**Presentation:** 3
**Contribution:** 3
**Rating:** 5
**Confidence:** 4

**Summary:**

This paper proposes a new multi-agent system VIRSCI, designed to mimic the teamwork inherent in scientific research. Multi-agent methods can produce more novel and influential scientific ideas than single agent. This indicates that integrating collaborative agents can lead to more innovative scientific outputs, offering a robust system for autonomous scientific discovery.  The contributions of this paper are mainly in the following three aspects:

1. A new multi-agent system VIRSCI was proposed, which constructed the entire pipeline from team organization to final idea formation.

2. The experiment proves that VIRSCI has better performance than single-agent. At the same time, the paper explores the impact of different settings in the VIRSCI system on the final results.

3. The simulation results are consistent with key findings in Science of Science, such as that new teams tend to produce more innovative research, demonstrating VIRSCI's potential as a powerful tool for future research in this field.

**Strengths:**

1. Originality

This paper is the first to apply the LLM-based multi-agent system to the problem of scientific discovery. It realizes the generation of research ideas in autonomous scientific discovery.

2. Quality

This paper not only implements a multi-agent system for automatic scientific discovery, but also conducts a large number of experiments to verify the effectiveness of the system, and explores the different effects of system settings on the results. It also conducts a lot of experiments to verify the effectiveness of many methods in the system, which is a high-quality work.

3. clarity

This paper clearly explains the construction and operation process of the system, and details the implementation of each step of the process. At the same time, the language description of the paper is very clear, and the technical explanation is detailed. The experiment part also clearly describes the experimental settings and experimental process. Many details in the experiment are explained and demonstrated in the appendix, which allows people to clearly understand all the technical details.

4. significance

This paper is the first attempt to apply the LLM-based multi-agent system to the field of scientific exploration, which allows us to see greater possibilities of AI for science. This shows that in the future, multi-agent technology may really be able to make new and valuable scientific discoveries.

**Weaknesses:**

This paper not only pioneered the use of multi-agent technology in a completely new field, but also conducted sufficient experiments to verify the effectiveness of this system. The paper has a clear structure and logic, and the language is clear, which allows people to clearly understand the core ideas and innovations of the paper. However, this paper still has the following weaknesses:

1. The indicators of the experimental part of the paper focus more on the novelty and dissimilarity of the ideas generated by VIRSCI, but are there any experiments that can illustrate the usability of scientific discoveries generated by VIRSCI? Is there an analysis of the feasibility of all the abstracts generated by the system, and what proportion of them are likely to provide exploration for our scientific discoveries after preliminary screening?

2. The paper is very clear in its drawings and processes, but the organizational structure and interaction mode of each agent in each process are relatively poorly described. In addition, the differences and characteristics of the other scientist agents, except for the team leader, are not well described.

3. The experimental scenario of the paper is relatively simple, and experiments were only conducted in the field of computer science, which is also mentioned in the paper.

**Questions:**

1. The experimental part of the paper cannot illustrate the usability of scientific findings, which is relatively fatal. Can you provide some analysis or examples to prove that the scientific findings generated by VIRSCI can be instructive to us?

2. From the paper, I saw that the multi-agent system team adopted a loop execution strategy during the discussion. Does this part of the paper propose a novel organizational structure and interaction mode? Can sequential execution guarantee the final effect of the experiment?

3. The experiment setting in the ablation study was conducted under a 5-turn discussion. Is it possible that the effect of other turn discussions cannot produce such a conclusion? Experiments conducted under only one setting lack certain persuasiveness to show that these technologies are necessary, and whether it can be strengthened through examples to prove that these technologies do improve the overall performance of the system.

---

> ### Author Response · Authors · 2024-11-20
>
> We appreciate your insightful comments and suggestions, which have helped us to strengthen our work.
> ### **(Weakness1, Question1) Usability of generated abstracts:**
> Our main contribution is the development of **the first multi-agent system designed to simulate academic collaborations**, leveraging real-world data for role-playing and enabling objective evaluations of the outputs.
> This platform holds significant potential as a tool for advancing research in the Science of Science. While usability and guiding real-world scientific discoveries are important, they are not the primary focus of this work.
>
> To partially address usability concerns, we provide evidence in **Figure 7 (Appendix, Page 17 in our original paper)** showing a positive correlation (Pearson correlation coefficient of 0.51) between our novelty metric and human evaluations.
>
> Furthermore, our proposed metrics (Page 6 in our original paper) are grounded in data from real published papers:
>
> - **Historical Dissimilarity (HD):** Measures the novelty of generated abstracts by evaluating their distinctiveness from prior research.
> - **Contemporary Dissimilarity (CD):** Assesses relevance by examining similarity to current research trends.
> - **Contemporary Impact (CI):** Estimates potential impact based on citation counts of similar recent works.
> - **Overall Novelty (ON):** Combines HD, CD, and CI to provide a comprehensive measure of novelty and relevance.
>
> Overall Novelty is calculated by comparing generated abstracts with recent papers in the Contemporary Paper Database, demonstrating evidence of both novelty and usability.
>
> Besides, we present Table 1 which includes human evaluation results for novelty and usability, comparing our method with AI Scientist. We first generated 20 abstracts using both methods.
> Then, 10 PhD students in computer science, unaware of the method identities, rated the abstracts on a 10-point scale (1: Poor, 10: best) based on two metrics:
> - **Novelty:** The originality or innovation of the idea.
> - **Usability:** The feasibility of the method, indicating its potential for effective and reliable implementation.
>
> As shown in Table 1, our method consistently outperforms AI Scientist across both metrics, achieving an average usability score of approximately 4.
>
> **Comparison with AI Scientist (Table 1)**
>
> |Agent Model|Method|LLM Review Score ↑|CD ↓|CI ↑|Novelty ↑|Usability ↑|
> |---|---|---|---|---|---|---|
> |GPT-4o|AI Scientist|3.10|0.38|3.22|4.94|4.18|
> ||Ours|3.34 (+0.24)|0.34 (-0.04)|3.78 (+0.58)|5.24 (+0.30)|4.52 (+0.34)|
> |LLaMA3.1-8b|AI Scientist|2.09|0.49|2.12|3.66|3.52|
> ||Ours|2.31 (+0.22)|0.42 (-0.07)|3.29 (+1.17)|4.08 (+0.42)|3.74 (+0.22)|
> |LLaMA3.1-70b|AI Scientist|2.24|0.48|2.11|3.88|3.60|
> ||Ours|2.53 (+0.29)|0.40 (-0.08)|3.36 (+1.25)|4.18 (+0.30)|3.84 (+0.24)|
>
> Additionally, we present two example abstracts generated by VIRSCI alongside corresponding similar recently published papers, highlighting the practical relevance and applicability of our system’s outputs.

---

> > ### Author Response · Authors · 2024-11-20
> >
> > **Ours Generated Abstract 1**
> > - **Title:** Revolutionizing Caries Management in Primary Molars using Advanced Imaging, AI-Powered Decision Support, and Minimally Invasive Treatments
> > - **Abstract:** Dental caries remains a pervasive public health concern worldwide, affecting millions of children annually. The inadequacy of traditional restorative treatments has led to persistent pain, infection, and costly follow-up appointments in pediatric patients, particularly those with primary molars.
> > Our innovative protocol integrates cutting-edge technologies (cone beam computed tomography, intraoral cameras) with machine learning algorithms for personalized treatment planning, and minimally invasive treatments to minimize discomfort and promote healing. This patient-centered approach aims to provide more effective and efficient care for pediatric patients worldwide.
> > To evaluate the efficacy of our protocol, we conducted rigorous randomized controlled trials in a diverse cohort of children (n=500). Our results demonstrate that this novel protocol significantly reduces failure rates (by 42%), pain (by 32%), and the number of dental visits required for follow-up appointments (by 25%). Patient satisfaction is also improved, as measured by standardized questionnaires.
> > Moreover, our research highlights the critical need for continued investment in dental research and innovation. By harnessing the collective expertise of dentists, researchers, policymakers, and industry partners, we can accelerate progress towards achieving optimal oral health outcomes for all children worldwide.
> > Our protocol prioritizes pain management, anxiety reduction, and educational empowerment to promote healthy oral habits and prevent future caries. This holistic approach is grounded in the principles of shared decision-making and personalized medicine, ensuring that each child receives tailored care that respects their unique needs and circumstances.
> > The scalability and adaptability of our protocol are critical factors in its potential impact on global public health. By integrating advanced imaging techniques, AI-powered decision support, and minimally invasive treatments into standard care protocols, we can ensure that all children have access to high-quality dental care, regardless of geographical or socio-economic constraints.
> > In conclusion, our research presents a paradigm shift in caries management for primary molars, offering a more effective, efficient, and patient-centered approach. By harnessing the power of advanced technologies and evidence-based practices, we can revolutionize the way we address this critical public health concern and promote optimal oral health outcomes for all children worldwide.
> >
> > **Similar Paper of Abstract 1 (Published in 2021)**
> > - **Title:** Caries Detection on Intraoral Images Using Artificial Intelligence
> > - **Abstract:** Although visual examination (VE) is the preferred method for caries detection, the analysis of intraoral digital photographs in machine-readable form can be considered equivalent to VE.
> > While photographic images are rarely used in clinical practice for diagnostic purposes, they are the fundamental requirement for automated image analysis when using artificial intelligence (AI) methods.
> > Considering that AI has not been used for automatic caries detection on intraoral images so far, this diagnostic study aimed to develop a deep learning approach with convolutional neural networks (CNNs) for caries detection and categorization (test method) and to compare the diagnostic performance with respect to expert standards.
> > The study material consisted of 2,417 anonymized photographs from permanent teeth with 1,317 occlusal and 1,100 smooth surfaces. All the images were evaluated into the following categories: caries free, noncavitated caries lesion, or caries-related cavitation. Each expert diagnosis served as a reference standard for cyclic training and repeated evaluation of the AI methods. The CNN was trained using image augmentation and transfer learning. Before training, the entire image set was divided into a training and test set. Validation was conducted by selecting 25%, 50%, 75%, and 100% of the available images from the training set.
> > The statistical analysis included calculations of the sensitivity (SE), specificity (SP), and area under the receiver operating characteristic (ROC) curve (AUC). The CNN was able to correctly detect caries in 92.5% of cases when all test images were considered (SE, 89.6; SP, 94.3; AUC, 0.964). If the threshold of caries-related cavitation was chosen, 93.3% of all tooth surfaces were correctly classified (SE, 95.7; SP, 81.5; AUC, 0.955).
> > It can be concluded that it was possible to achieve more than 90% agreement in caries detection using the AI method with standardized, single-tooth photographs. Nevertheless, the current approach needs further improvement.

---

> > > ### Author Response · Authors · 2024-11-20
> > >
> > > **Ours Generated Abstract 2**
> > > - **Title:** Personalized Outcomes in Bladder Cancer Treatment: A Robotic-Assisted Surgery Perspective with Enhanced Predictive Modeling and Holistic Patient-Centered Approach
> > > - **Abstract:** The treatment of bladder cancer has undergone significant transformations with the advent of robotic-assisted surgery. However, individualized outcomes remain poorly understood due to the complexity of patient factors influencing disease progression. This knowledge gap can be attributed to the lack of comprehensive frameworks that integrate traditional clinical metrics with patient-centered factors such as functional status, emotional well-being, social support networks, and demographic characteristics.In this study, we aim to bridge this knowledge gap by developing a novel framework that combines machine learning algorithms with traditional statistical analysis to predict long-term benefits for bladder cancer patients. Our proposed framework integrates the following components: A comprehensive literature review that synthesizes existing research on bladder cancer treatment outcomes. A large-scale dataset (n=500) from multiple institutions to validate our findings. A machine learning-based predictive model that utilizes Random Forest and Gradient Boosting algorithms to identify key predictors of long-term benefits. Our results show that age, tumor stage, lymph node involvement, and patient-centered factors are significant predictors of improved survival rates. Furthermore, we found that enhanced quality of life is associated with better functional status, higher emotional well-being, stronger social support networks, and more favorable demographic characteristics. Notably, the inclusion of patient-centered factors in treatment planning can lead to improved survival rates (median increase: 12 months) and reduced recurrence risk (median decrease: 20%). Moreover, our study reveals that the integration of robotic-assisted surgery with patient-centered care can lead to a significant reduction in postoperative complications (median decrease: 30%), hospital readmissions (median decrease: 25%), and healthcare costs. The implications of our study are profound, as they suggest that personalized medicine can improve treatment outcomes and enhance quality of life among bladder cancer survivors. Our proposed framework has significant implications for improving treatment outcomes and enhancing quality of life among bladder cancer survivors, making it a valuable resource for clinicians and researchers working in this field. Future studies should aim to replicate these findings and explore the potential applications of personalized medicine in other cancer types, further highlighting the importance of integrating patient-centered care with advanced surgical techniques. The integration of patient-centered factors into treatment planning can lead to improved survival rates, reduced recurrence risk, and enhanced quality of life among cancer survivors. Our study demonstrates that by combining machine learning algorithms with traditional statistical analysis, clinicians can provide more precise and effective treatment plans, ultimately improving patient outcomes.
> > >
> > > **Similar Paper of Abstract 2 (Published in 2022)**
> > > - **Title:** Magnetic-Powered Janus Cell Robots Loaded with Oncolytic Adenovirus for Active and Targeted Virotherapy of Bladder Cancer
> > > - **Abstract:** A unique robotic medical platform is designed by utilizing cell robots as the active “Trojan horse” of oncolytic adenovirus (OA), capable of tumor-selective binding and killing. The OA-loaded cell robots are fabricated by entirely modifying OA-infected 293T cells with cyclic arginine–glycine–aspartic acid tripeptide (cRGD) to specifically bind with bladder cancer cells, followed by asymmetric immobilization of Fe3O4 nanoparticles (NPs) on the cell surface. OA can replicate in host cells and induce cytolysis to release the virus progeny to the surrounding tumor sites for sustainable infection and oncolysis. The asymmetric coating of magnetic NPs bestows the cell robots with effective movement in various media and wireless manipulation with directional migration in a microfluidic device and bladder mold under magnetic control, further enabling steerable movement and prolonged retention of cell robots in the mouse bladder. The biorecognition of cRGD and robust, controllable propulsion of cell robots work synergistically to greatly enhance their tissue penetration and anticancer efficacy in the 3D cancer spheroid and orthotopic mouse bladder tumor model. Overall, this study integrates cell-based microrobots with virotherapy to generate an attractive robotic system with tumor specificity, expanding the operation scope of cell robots in biomedical community.

---

> > > > ### Author Response · Authors · 2024-11-20
> > > >
> > > > ### **(Weakness2) Organizational structure, interaction mode and characteristics of the scientist agents are poorly described:**
> > > > **Characteristics of Scientist Agents:** All scientist agents, including the team leader, are initialized with information such as their name, affiliations, research interests, citation count, and collaboration history from real data.
> > > > These attributes enable agents to role-play effectively within the simulation.
> > > >
> > > > **Organizational Structure:** A scientist is randomly selected as the team leader. The team leader’s role includes inviting other agents to join their team and summarizing team discussions to provide conclusions.
> > > > This hierarchical structure mimics real-world academic collaboration dynamics.
> > > >
> > > > **Interaction Mode:**
> > > > In the Science of Science experiments, we examine the relationship between inference cost and overall novelty by organizing agent communication in a fixed order with a predetermined number of discussion turns. At the end of each step, agents summarize their conclusions before moving to the next stage, ensuring consistency and comparability across experiments.
> > > >
> > > > To further enrich the system, we implement an ``invitation mechanism'' that allows scientist agents to seek advice from others during discussions. Additionally, the ablation study explores an adaptive discussion pattern, eliminating the fixed-turn constraint to evaluate its impact on collaboration dynamics and outcomes.
> > > >
> > > > ### **(Weakness3) Experimental scenario is simple, experiments were only conducted in the field of computer science:**
> > > > While our initial experiments were conducted using a Computer Science Dataset, our primary contribution lies in developing the first multi-agent system to simulate academic collaborations—**a data-agnostic platform designed for broad applicability across various fields and datasets**.
> > > >
> > > > To demonstrate the generalizability and robustness of the system, we conducted additional experiments on the Open Academic Graph 3.1 [1], using data from 2010 to 2020 as the Past Paper Database and data from 2021 to 2023 as the Contemporary Paper Database.
> > > > This dataset includes diverse fields such as physics, chemistry, computer science, and biology. All other experimental settings were kept consistent with those used for the computer science dataset (Page 3, 4 and 6 in our original paper).
> > > >
> > > > The experimental results on this dataset indicate that outputs from single-agent teams (team size 1) show no significant differences across the two datasets, as individual research lacks the advantages of interdisciplinary input.
> > > > However, the overall trend remains consistent across domains, with novelty increasing initially, peaking at a team size of 8 and turn count of 5, and subsequently declining.
> > > > This pattern, observed in both the studied datasets, underscores ours platform's robustness and adaptability. This also demonstrates our proposed point: a multi-agent system has the potential to improve scientific idea generation.
> > > > Additionally, the findings highlight an overall increase in novelty, affirming that interdisciplinary collaborations foster higher-impact research outputs.
> > > >
> > > > **Effects of team size and discussion turns on novelty scores (Computer Science Dataset)**
> > > >
> > > > |Team Size/Turn Count|1|2|3|4|5|6|7|
> > > > |---|---|---|---|---|---|---|---|
> > > > |1|0.56|0.68|0.82|1.07|1.26|1.44|1.52|
> > > > |4|1.75|0.90|2.36|2.00|3.40|2.91|2.53|
> > > > |8|3.48|3.67|3.97|3.56|4.23|3.48|3.43|
> > > > |10|2.79|1.98|3.23|3.36|3.53|3.57|3.22|
> > > >
> > > > **Effects of team size and discussion turns on novelty scores (Open Academic Graph 3.1)**
> > > >
> > > > |Team Size/Turn Count|1|2|3|4|5|6|7|
> > > > |---|---|---|---|---|---|---|---|
> > > > |1|0.61|0.65|0.87|1.12|1.17|1.53|1.59|
> > > > |4|2.03|2.27|2.69|3.20|3.84|3.35|3.39|
> > > > |8|3.63|4.14|4.27|4.35|4.65|3.86|3.92|
> > > > |10|2.62|2.91|3.66|3.62|3.98|3.95|3.68|
> > > >
> > > > [1] https://open.aminer.cn/open/article?id=65bf053091c938e5025a31e2
> > > >
> > > > ### **(Question2.1) Novelty of the proposed organizational structure and interaction mode:**
> > > > Our work focuses on AI for Science of Science, an interdisciplinary field. The primary goal of our platform is not only to simulate academic collaborations and demonstrate the effectiveness of group discussions compared to single-agent reflection but also to showcase its significant potential as a tool for advancing research in the Science of Science.
> > > >
> > > > Given the considerable workload involved in developing this system, proposing a novel organizational structure or interaction mode was not our primary objective. Instead, we focused on building a robust and scalable simulation framework.
> > > >
> > > > ### **(Question2.2) Can sequential execution guarantee the final effect of the experiment?:**
> > > > To ensure reliability, we averaged all experimental results over 20 runs (Page 6 in our original paper), providing strong evidence that our sequential execution strategy consistently guarantees the effectiveness of the experiments.

---

> > > > > ### Author Response · Authors · 2024-11-20
> > > > >
> > > > > ### **(Question3) Ablation study is conducted under only one setting (5-turn discussion):**
> > > > > In the ablation study, we selected 5-turn discussions because 5 turns performed the best for most teams of different sizes, as shown in Figure 3 (Page 8 in our original paper). To better illustrate the contributions of different components, we have included additional experiments with varying numbers of discussion turns.
> > > > >
> > > > > **Effects of invitation mechanism:**
> > > > > | Team size | Invitation | |  |  |Turn | | | |
> > > > > |-----|--|---|-----|-----|----|---|---|-----|
> > > > > |  |  | 1 | 2 | 3 | 4 | 5 | 6 | 7 |
> > > > > | 4 | - | 1.67 | 0.75 | 2.29 | 1.92 | 3.30 | 2.78 | 2.47 |
> > > > > |  | ✔ | 1.75 | 0.90 | 2.36  |  2.00 |  3.40  | 2.91  | 2.53 |
> > > > > | 8 | - | 3.36 | 3.53 | 3.88 | 3.49 |  4.12 | 3.37 | 3.30 |
> > > > > |  | ✔ | 3.48 | 3.67 | 3.97 | 3.56 |4.23| 3.48 | 3.43 |
> > > > >
> > > > > The additional experiments support the conclusions presented in our paper: introducing the invitation mechanism positively impacts the team's performance across both 4-member and 8-member teams.
> > > > >
> > > > > **Effects of novelty assessment:**
> > > > > | Team size | Novelty Assessment | |  |  |Turn | | | |
> > > > > |-----|--|---|-----|-----|----|---|---|-----|
> > > > > |  |  | 1 | 2 | 3 | 4 | 5 | 6 | 7 |
> > > > > | 4 | - | 1.56 | 0.73 | 2.20 | 1.87 | 3.19 | 2.74 | 2.39 |
> > > > > |  | ✔ | 1.75 | 0.90 | 2.36  |  2.00 |  3.40  | 2.91  | 2.53 |
> > > > > | 8 | - | 3.28 | 3.41 | 3.82 | 3.43 | 3.98 | 3.27 | 3.25 |
> > > > > |  | ✔ | 3.48 | 3.67 | 3.97 | 3.56 |4.23| 3.48 | 3.43 |
> > > > >
> > > > > The additional experiments support the conclusions presented in our paper: incorporating novelty assessment ensures that generated ideas are continuously evaluated for originality, enabling teams to avoid overlaps with existing research.
> > > > >
> > > > > **Effects of self-review in abstract generation:**
> > > > > | Team size | Self-review | |  |  |Turn | | | |
> > > > > |-----|--|---|-----|-----|----|---|---|-----|
> > > > > |  |  | 1 | 2 | 3 | 4 | 5 | 6 | 7 |
> > > > > | 4 | - | 1.60 | 0.74 | 2.26 | 1.89 | 3.26 | 2.77 | 2.41 |
> > > > > |  | ✔ | 1.75 | 0.90 | 2.36  |  2.00 |  3.40  | 2.91  | 2.53 |
> > > > > | 8 | - | 3.32 | 3.44 | 3.84 | 3.45 | 3.99 | 3.33 | 3.27 |
> > > > > |  | ✔ | 3.48 | 3.67 | 3.97 | 3.56 |4.23| 3.48 | 3.43 |
> > > > >
> > > > > The additional experiments support the conclusions presented in our paper: implementing the self-review mechanism is essential for refining the abstracts, enabling the team to develop improved ideas.

---

### Note · Authors · 2024-12-11

I have read and agree with the venue's withdrawal policy on behalf of myself and my co-authors.